# Recognition and reprogramming of E3 ubiquitin ligase surfaces by α-helical peptides

Olena S. Tokareva[1,7], Kunhua Li [1,5,7], Tara L. Travaline [1], Ty M. Thomson [1], Jean-Marie Swiecicki [1,6], Mahmoud Moussa[1], Jessica D. Ramirez [1], Sean Litchman [1], Gregory L. Verdine [1,2,3,4] ✉ & John H. McGee [1] ✉

Molecules that induce novel interactions between proteins hold great promise for the study of biological systems and the development of therapeutics, but their discovery has been limited by the complexities of rationally designing interactions between three components, and because known binders to each protein are typically required to inform initial designs. Here, we report a general and rapid method for discovering α-helically constrained (Helicon) polypeptides that cooperatively induce the interaction between two target proteins without relying on previously known binders or an intrinsic affinity between the proteins. We show that Helicons are capable of binding every major class of E3 ubiquitin ligases, which are of great biological and therapeutic interest but remain largely intractable to targeting by small molecules. We then describe a phage-based screening method for discovering "trimerizer" Helicons, and apply it to reprogram E3s to cooperatively bind an enzyme (PPIA), a transcription factor (TEAD4), and a transcriptional coactivator (β-catenin).

Protein-protein interactions (PPIs) play a central role in nearly all biological processes, from the binding of protein or peptide ligands to their receptors, to the modulation of the activity and specificity of enzymes, to the scaffolding of signaling cascades and other functional complexes, all of which can become dysregulated in disease. As a consequence, there is considerable scientific and therapeutic interest in developing molecules that modulate PPI activity. Historically, efforts have largely focused on agents that disrupt PPIs[1], but considerable progress has been made in recent years to induce the formation of novel PPIs[2], particularly in the context of reprogramming E3 ubiquitin ligases to recognize novel substrate proteins and mark them for proteasomal degradation[3].

A key constraint facing the rational design of molecules that induce novel interactions between proteins is the typical requirement for known binding ligands to each, or a known interaction between the two proteins or close relatives, to serve as a starting point for designs. Because most proteins cannot be effectively bound by small molecules (the "druggability" problem[4]), this constraint significantly limits the proteins for which small molecule-based PPI-inducing agents can be developed. This limitation is particularly acute for E3 ubiquitin ligases, of which only a handful can be bound by small molecules.

In response to this limitation, there has been an increased focus on developing peptide-based solutions to modulate therapeutically relevant PPIs, given their ability to engage significantly larger surfaces than small molecules[5,6]. This size advantage of peptides is particularly relevant when considering the need to stabilize PPIs through cooperative high-affinity interactions[7], for instance where pre-organization of two components enhance binding of a third.

[1]FOG Pharmaceuticals Inc., Cambridge, MA, USA. [2]Department of Stem Cell and Regenerative Biology, Harvard University, Cambridge, MA, USA. [3]Department of Chemistry and Chemical Biology, Harvard, University, Cambridge, MA, USA. [4]Department of Molecular and Cellular Biology, Harvard University, Cambridge, MA, USA. [5]Present address: Kymera Therapeutics, Inc., Watertown, MA, USA. [6]Present address: Relay Therapeutics, Inc., Cambridge, MA, USA. [7]These authors contributed equally: Olena S. Tokareva, Kunhua Li. ✉e-mail: gregory_verdine@harvard.edu; jmcgee@fogpharma.com

Since peptides generally possess large numbers of exposed polar and charged groups, a critical challenge in using them to modulate the function of proteins and to engineer PPIs has been their delivery into cells. Over the past several decades, we and others have reported chemical approaches to reinforce the α-helical structure of peptides to increase their ability to cross cellular membranes[8–11], leading to the development of α-helically constrained (Helicon) peptides[12,13]. In addition to possessing increased stability and membrane permeability, Helicons are capable of binding large, flat protein surfaces that are inaccessible to targeting by small molecules, and Helicons have been developed to target a range of PPI targets in cells and in vivo[14–18].

We therefore hypothesized that Helicons, with their expanded surface area and ability to engage surfaces that small molecules cannot, present an ideal modality with which to develop agents that could induce novel PPIs based on cooperative binding events. Here, we report a general method for discovering Helicons that cooperatively lead to the interaction between two proteins, which we term "trimerizer" Helicons, without relying on known binders to either. We apply this method, which is accessible to laboratories with standard molecular biology equipment and access to a DNA sequencing core facility, to discover trimerizer Helicons that induce the binding of E3 ligases to target proteins for which they have no intrinsic affinity, thereby reprogramming their surface-recognition behavior.

There are an estimated 600 human E3s, grouped into four families, but only a small number of these have been successfully co-opted for targeted protein degradation (TPD)[19] applications using small molecules, called "molecular glues"[20,21], or "degraders" if they induce TPD. We recently introduced an unbiased high-throughput screening platform utilizing cysteine-stapled phage display to discover Helicons that can engage novel surfaces on difficult targets, including RNF31, a member of the RING-in-between-RING (RBR) E3 ligase family[22]. Here, we extend that work to identify Helicons that bind to eight additional E3 proteins from the three remaining E3 ligase families. We then developed a new screening approach that enables the direct discovery of cooperative, molecular glue-like binders, which we term "trimerizer" Helicons, that lead to ternary complex formation between E3 ligases and target proteins.

From the first set of high-throughput screens, we identified Helicons that bind to these E3s from a naive $10^8$-member phage display library of 14-mer Helicons. This resulted in the discovery of Helicons that bind members of the HECT family (WWP1 and WWP2), members of the Cullin-RING (CRL) multi-subunit E3 family consisting of Cullin proteins paired with their canonical adapter proteins (CUL1-FBXW7, CUL2-VHL, and CUL5-SOCS2), and RING/U-Box family members Murine double minute 2 (MDM2) and C-terminus of HSC70-interacting protein (CHIP/STUB1). Characterizing Helicon-E3 co-structures and mechanisms of action, we identified new binding sites for α-helices on the E3 surfaces, as well as potential probes of the disease-linked E3, WWP1[23,24], highlighting the generality of our approach against this therapeutically important target class.

These binders were then used to inform the generation of new, "focused" libraries of 20-mer Helicons, wherein the E3-binding residues were fixed and the remainder were randomized. By subsequently screening these focused libraries against target proteins in the presence or absence of the E3 presenter protein, we were able to directly identify trimerizer Helicons that cooperatively bind the targets only in the presence of the E3 "presenter" protein.

Using this platform, we discovered trimerizers that induce interactions between the E3 ubiquitin ligase CHIP and the peptidyl-prolyl cis-trans isomerase Cyclophilin A (PPIA), between CHIP and the TEA domain transcription factor TEAD, and between the E3 ubiquitin ligase MDM2 and the transcriptional coactivator β-catenin. Biochemical and biophysical assessment of the trimerizers derived from these screens confirmed their cooperative binding and their ability to promote protein-protein interactions, and x-ray co-crystal structures of two

trimerizers between MDM2 and β-catenin revealed the structural basis of interactions both between the Helicon and each protein and between the two proteins themselves. We anticipate that this method will prove useful for the discovery of Helicons and to identify the molecular recognition events that induce the interaction of proteins that have otherwise proven challenging to engage.

## Results

### A phage-based method for discovering trimerizer Helicons without prior binders or structural information

Our approach for discovering trimerizer Helicons involves two single-round phage screens, performed in succession, each using ~$10^8$-membered cysteine-stapled Helicon libraries (Fig. 1a). The first is a screen of a naive library (as described previously[22]) for the presenter protein. The hits from this screen are then used to design a focused phage library that is biased to bind the presenter, but remains diverse at residues that do not bind the presenter (Fig. 1b). The second screen is then performed using the focused library, this time screening for binding to the target protein. Importantly, this is done under two conditions: with the presenter protein present in solution (although not immobilized to the selection beads), and with the presenter absent. Two possible types of target-binding behavior can result: binding that occurs both in the presence and absence of presenter protein, and binding that occurs only in the presence of presenter. These latter, presenter-dependent binders are selected for synthesis and validation, since their presenter dependence suggests the desired cooperative binding mode to their target (Fig. 1c).

We have shown previously that highly conserved positions in the sequence cluster logos from naive screens serve as accurate indicators of which residues are directly involved in target binding[22]. We reasoned that this knowledge could be leveraged to design focused libraries purely from logos, wherein conserved positions in the logos for the presenter are fixed, and the non-conserved residues are diversified. Trimerizer library primers can be constructed accordingly, using a combination of defined, semi-degenerate, and fully degenerate codons to match the parent cluster logo as closely as possible. Crucially, since this approach does not require structural information for library design, libraries can be created immediately following the first screen, without the need for costly and time-consuming hit validation and structure determination.

### A screening campaign to discover E3-binding Helicons for the HECT, Cullin-RING, and RING/U-box families

We began by mapping the Helicon binding sites on a range of E3 ubiquitin ligase presenters using naive libraries. While E3s were considered largely undruggable until the last decade, the TPD toolbox is rapidly expanding, and includes several clinically validated molecules[25]. However, of the estimated 600 human E3s, only a select few have been routinely targeted by small-molecule molecular glue-like molecules[26]. These include molecules that co-opt the activity of Von Hippel-Lindau (VHL) and Cereblon (CRBN), which are members of the multi-subunit Cullin-RING (CRL) family, and of MDM2, RING-finger protein RNF4, and inhibitor of apoptosis protein (IAP), which are members of the single-subunit RING-finger/U-Box family. We sought to expand this toolbox significantly, including against E3 ligases with a range of substrate recruitment capabilities, substrate specificities, and tissue distribution profiles (Supplementary Fig. 1a), by screening for Helicons that target members of each of the four major E3 families.

We described the beginning of this campaign with the original report of our phage display-based screening platform[22], which included the discovery of cysteine-stapled Helicons that bind to the RING-in-between-RING (RBR) E3 ligase RNF31 (Supplementary Fig. 1b, c). In the current work, we report the results of screening against members of the other three large families of E3 ubiquitin ligases: the HECT, CRL, and RING/U-Box families.

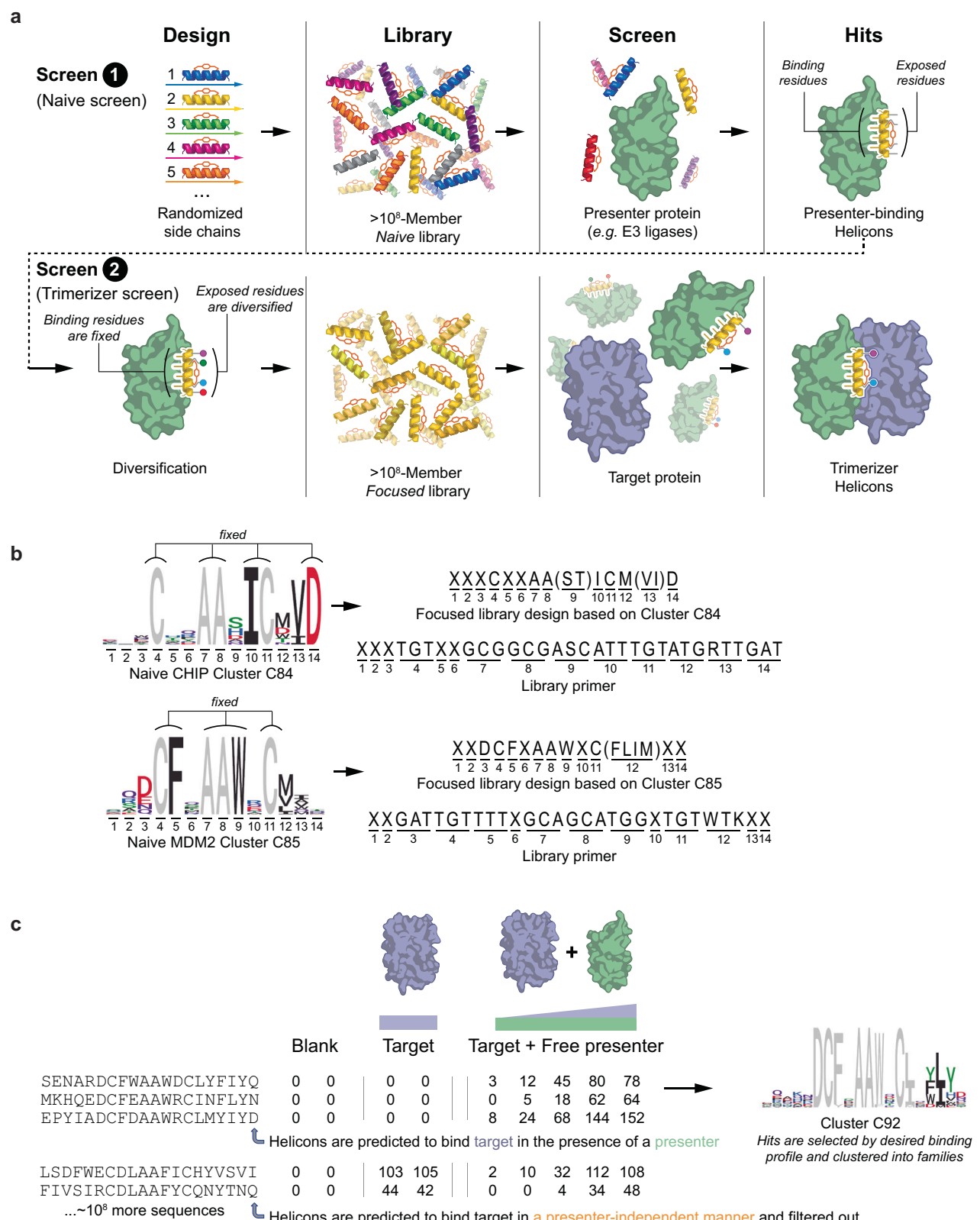

## Helicons that bind the HECT E3 family

First, we focused on the NEDD4-like E3 ligases, which make up approximately a third of the HECT family[27]. These include WWP1 and WWP2, which each consist of four tandem WW domains and a catalytic HECT domain at the C-terminus that transfers ubiquitin from a bound E2 first to itself and subsequently to the target substrate (Fig. 2a). WWP1 induces the ubiquitination of several

tumor suppressor proteins including p53 and PTEN[23,28], and its increased expression in some human cancers promotes cell proliferation, migration, and invasion, making it an attractive therapeutic target[28]. For instance, it has been reported that inhibition of the MYC-WWP1 axis involved in PTEN regulation using the natural product indole-3-carbinol can suppress tumors by reactivating PTEN[23].

**Fig. 1 | A general method for the discovery of trimerizer Helicons that induce a novel interaction between two target proteins. a** In the first of two high-throughput screens (Screen 1), Helicons that bind to the presenter E3 proteins (green) are identified from a naive $10^8$-member phage display library, as described previously[22] and as in Figs. 2 and 3. Screen 2 begins by designing and generating a focused Helicon library based on hits from Screen 1. Helicon residues involved in E3 binding are determined by the prominence of the amino acid letters in the cluster logo, and from Helicon co-structures with the E3 if available. These residues are fixed and the remaining residues are randomized in focused library primers that are used to build a library of 20-mer Helicons with a diversity of ~$10^8$ members. Screening for target binders from the diversified library is done in the presence or absence of the E3 presenter protein to identify trimerizer hits that bind the target in a presenter-dependent manner. **b** Two examples of focused library designs. In the case of CHIP cluster C84, residues 10, 12, and 14 are fixed, while the remaining

residues are either partially randomized (residues 9 and 13) since they were defined by two to five favored residues in the logo, or fully randomized (residues 1–3, 5, 6, plus six added residues flanking the 14-mer core). Amplification primers contain partially degenerate codons (X) or semi-degenerate codons (e.g. RTT for valine or isoleucine at position 13 in the CHIP focused library) for library production (Supplementary Table 2). The four fixed cysteine and alanine residues from the naive library design remain fixed in the focused library. Similar design logic is used for the MDM2 cluster C85. **c** Hits selected from focused phage libraries were screened for binding to the target (blue) in the presence or absence of presenter (green). Quantitation of target binding (illustrative tabular values) is used to identify trimerizer Helicons that bind the target only in the presence of the presenter (i.e., cooperatively), and these are clustered based on sequence (top). Helicons that bind the target even in the absence of presenter (i.e., non-cooperatively) are ignored (bottom).

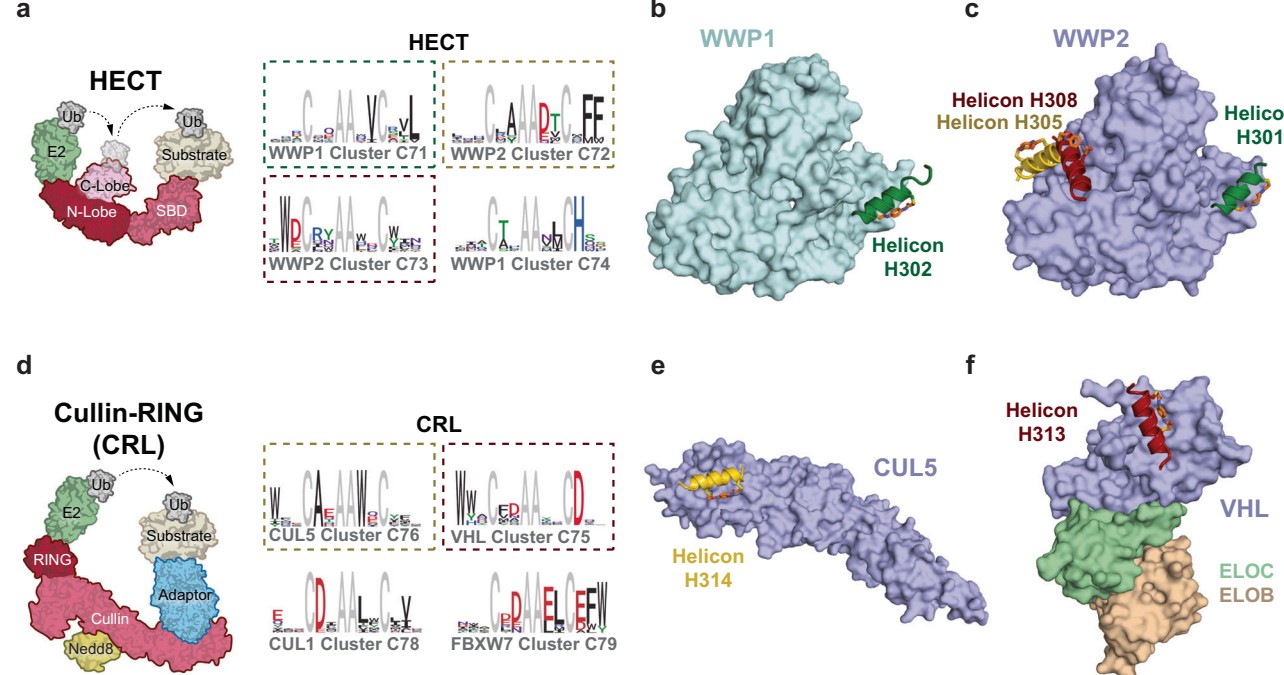

**Fig. 2 | Discovery of Helicon binding sites for members of the HECT and CRL E3 ligase families. a** A prototypical HECT E3 ligase where two lobes of the ~40 kDa HECT domain are flexibly tethered. The E2-binding surface is on the larger N-lobe, while the active-site cysteine is on the C-lobe. Ubiquitin (Ub)-loaded E2 brings the HECT and E2 cysteines into close proximity for ultimate Ub transfer to protein targets bound to the substrate-binding domain (SBD) of the E3 ligase. Representative clusters discovered by screening a Helicon library against WWP-family protein constructs that either stabilize the autoinhibited, inactive state (WWP1$^{WW-HECT}$, residues 379–922) or that can adopt both active and inactive conformations (WWP1$^{HECT}$, residues 546–917 and WWP2$^{HECT}$, residues 492–865, respectively) are depicted as logos, with gray indicating residues that were fixed for all library members. **b** Co-crystal structures of Helicon H302 with WWP1$^{HECT}$ (PDB: 8EI4) and **c** overlaid structures of H301 (PDB: 8EI5), H308 (PDB: 8EI8), and H305

(PDB: 8EI6) with WWP2$^{HECT}$, solved at 2.43–3.60 Å resolution (see Supplementary Data 2). **d** A prototypical multi-subunit Cullin-RING (CRL) E3 ligase with an E2-binding RING domain, shown with a single adapter subunit (as in CUL3-scaffolded CRLs). The adapter component of CUL1-, CUL2-, CUL4-, CUL5-, and CUL7-scaffolded CRLs consist additionally of a receptor subunit (e.g., VHL plus ELOBC for CUL2). Representative clusters derived from screening a Helicon library against CUL5 (residues 1–386), VHL-ELOBC (VHL residues 54–213 and ELOB residues 1–104, and ELOC residues 17–112), CUL1 (residues 15–410), and FBXW7-SKP1 (FBXW7 residues 263–706 and full-length SKP1 residues 1–163) are depicted as logos as in **a**. **e** Co-crystal structure of H314 bound to the N-terminal domain of CUL5 (residues 8–384, CUL5$^{NTD}$) (PDB: 8EI2). **f** H313 binding to VHL in the VHL-ELOBC complex (PDB: 8EI3). Composite omit maps of these Helicons are shown in Supplementary Fig. 7.

We began by designing three HECT-containing protein constructs: a construct that stabilizes the autoinhibited, inactive state of WWP1 (WWP1$^{WW-HECT}$), and WWP1/2 constructs that are able to adopt both the active and inactive conformations (WWP1$^{HECT}$ and WWP2$^{HECT}$). The WWP1$^{WW-HECT}$ construct that locks WWP1 in an inactive state consists of WW domains 2–4, an inhibitory linker region that connects WW domains 2 and 3, and the HECT domain[29]. We screened these three recombinant proteins in vitro in parallel against a naive phage library that displays ~$10^8$ 14-mer Helicons, and used hierarchical statistical clustering to group the binders into families of related sequences, represented as pharmacophore logos (Fig. 2a, Supplementary Fig. 1d).

We identified several binding profiles (Supplementary Fig. 1d, Supplementary Table 1), with four of them described below. The first profile (Group 1) represents Helicons such as those in cluster C71 (C71) that are pan-WWP-binders – binding indiscriminately to inactive and active forms of both HECT domains, which share ~70% sequence identity. Group 2 Helicons, the pan-HECT group including C73, bind both HECT domains, while Group 3 Helicons, the WWP2-HECT group including C72, bind only the WWP2 HECT domain (WWP2$^{HECT}$). The Group 4 Helicons, represented by C74, bind to WWP1$^{WW-HECT}$ but not WWP1$^{HECT}$, presumably binding via the WW domain.

We synthesized, cysteine stapled, and biochemically validated selected Helicons from representative clusters from each group. Most of the phage hits bound the proteins with µM affinity, as assessed by surface plasmon resonance (SPR) (Supplementary Fig. 1d, Supplementary Data 1). The most potent hit from this single-round screen, H306, had a binding affinity ($K_D$) of 390 nM, against WWP2[HECT]. We found that individual Helicons from all three HECT-binding Groups (1–3) could modestly inhibit the auto-ubiquitylation activity of the isolated WWP2[HECT] (Supplementary Fig. 1e).

We next solved the crystal structures of five Helicons, four in complex with WWP2[HECT] representing all three groups of HECT-binding clusters, and one from Group 1 with WWP1[HECT] (Fig. 2b, c, Supplementary Figs. 1f, g, Supplementary Data 1). These structures revealed three distinct Helicon binding sites on WWP2[HECT]. In all of the structures, we found that the HECT domains adopted similarly closed (inhibited) conformations[29]. The WWP2 proteins in the co-structures with H301, H305, and H308 share a low root-mean-square deviation (RMSD = 0.5–0.8 Å) and are shown overlaid (Fig. 2c). Consistent with their ability to impact the catalytic activity of WWP2[HECT], H301 binds near the E2 interface, while H305 and H308 may restrict the conformational change involved in activating this domain. As shown in our previous work[22], the prominence of the individual amino acid letters in the logo again correlated well with their role in direct binding to the target protein.

Although they belonged to different clusters, the Group 2 Helicons H304 and H308 bound WWP2[HECT] using similar residues, interacting with N- and C-lobes of the HECT domain and overlapping the binding site of WWP2's linker domain, which is necessary for its auto-inhibition[29] (Fig. 2c, Supplementary Fig. 1f, g). While the WWP2 linker region is mostly helical, it is linear where it overlaps the H308-binding site. Group 3 Helicon H305 binds a similar location near the interface of the N- and C-lobes of WWP2[HECT], with binding residues predominantly in contact with the C-lobe.

Group 1 Helicons H302 (WWP1[HECT]-binding) and H301 (WWP2[HECT]-binding) showed similar binding modes as each other, making contact only with the N-lobe of WWP1[HECT] (Fig. 2b) or WWP2[HECT], respectively. (Fig. 2c, Supplementary Fig. 1f). The HECT domain N-lobe has been shown to interact with both ubiquitin[30] and E2 ubiquitin ligases[31]. The Group 1 Helicons engage this same E2-binding site within WWP2[HECT] (Supplementary Fig. 1g). A similar binding site on HECT-family E3 ligases has been described for bicyclic peptides that act as competitive inhibitors of the E3 catalytic activity via disrupting interactions with cognate E2 ligases[32]. Since the E2-binding site is not strictly conserved among HECT family E3 ligases[33], it is theoretically possible to develop HECT-selective, E2-competing Helicons.

## Helicons that bind the Cullin-RING E3 family

Having established that our screening platform can successfully discover novel binding sites and Helicon binders against the HECT family of E3 ligases, we moved on to the larger Cullin-RING (CRL) family. Just as the HECT family has been shown to be coopted by viral oncoproteins, exemplified by human papilloma virus E6 which binds to host E3 E6AP to induce degradation of the tumor suppressor p53 (refs. 34,35), there are numerous examples of viral proteins that can recruit and rewire the substrate specificities of host CRLs to induce degradation of host immunity proteins[36]. We sought to determine whether Helicons could be discovered that would modulate CRL structure and function, and inform the development of tools for inhibiting and promoting PPIs.

CRL is the largest E3 family, and is unique in that its members are modular[37] – Cullin (CUL) proteins form the central scaffold that recruits an E2 enzyme via a RING-Box protein (typically RBX1 or RBX2) to its N-terminus and a CUL-specific adapter that bridges the C-terminus of the CUL protein to the substrate protein (Fig. 2d). In the case of CRLs scaffolded by CUL1 and CUL7 (CRL1 and CRL7), the adapter is a sub-complex called SCF consisting of SKP1 and an F-box protein, while the adapters scaffolded by CUL2, CUL3, CUL4A/4B, and CUL5 are variable substrate recruitment subunits. VHL (CRL2) and CRBN (CRL4) are among the most prominent members of the CRL class, and given the availability of small molecule ligands that recruit them, are the targets of most PROteolysis Targeting Chimeras (PRO-TACs) – heterobifunctional molecules for TPD generally consisting of two small molecule ligands that bind separately to the E3 and the target protein, connected by a linker[38].

We performed screens to find Helicons that bind to CRLs consisting of Cullin proteins paired with their canonical adapter proteins. First, to find Helicons that interact specifically with CRL1, CRL2, or CRL5, we purified the N-terminal domains of CUL1, CUL2, and CUL5 and screened them in parallel with the counter-target CUL4B. We also included the corresponding CRL substrate-recognition adapters, including VHL and the ElonginB/ElonginC complex (ELOBC) for CUL2. We identified Helicon hits forming VHL-ELOBC-specific clusters and CUL5-specific clusters (Fig. 2d, Supplementary Data 1). We also identified hits for CUL1, CUL2, FBXW7-SKP1, and SOCS2-ELOBC (Supplementary Data 1). None of the CRL-binding Helicons we identified contained substrate motifs known to be recognized by this family of E3 ligases, such as the C-terminal degron RxxG[39].

We next used biochemical and biophysical approaches to confirm that the clusters and hits we discovered were target-specific and bind to biologically relevant sites. Among the VHL-ELOBC-specific binding clusters, we identified Helicon H313 (C75) that binds VHL-ELOBC, but does not compete with the previously reported fluorescent VHL-binding probe, HXC78 (ref. 40), and it did not bind SOCS2-ELOBC that acted as a counter-target for VHL-ELOBC in the CRL phage screens (Supplementary Data 1). Using SPR, we confirmed the specificity of H313 for VHL-ELOBC over SOCS2-ELOBC and measured the $K_D$ for the former to be 4.1 µM (Supplementary Fig. 2a). We also used a fluorescence polarization (FP) competition assay to show that H313 does not inhibit the HXC78 interaction with VHL-ELOBC, while the VHL ligand VH298, which is related to HXC78 (ref. 41), does (Supplementary Fig. 2b). Among the CUL5-binding Helicons, H314 from C76 bound with a $K_D$ of 530 nM as assessed by SPR (Supplementary Fig. 2c). H314 also disrupts the SOCS2-ELOBC:CUL5 interaction in a competition SPR (ABA) binding assay, potentially by competing for the adapter-binding site of CUL5 (Supplementary Fig. 2c).

We further characterized by x-ray crystallography Helicons derived from these screens. We solved the structure of H314 with the CUL5 N-terminal domain (CUL5[NTD]) at ~2.8 Å resolution (Fig. 2e), H313 in complex with VHL-ELOBC at ~3.5 Å resolution (Fig. 2f, Supplementary Fig. 2d), and CUL4B-specific H316 from C77 in complex with the N-terminal domain of CUL4B (CUL4B[NTD]) at ~2.9 Å resolution (Supplementary Fig. 2e). Notably, the co-structure with CUL4B[NTD] reveals binding to a site that is not biologically relevant since it is not available in the native context of the CUL4B C-terminal domain (CUL4B[CTD]). This result highlights the importance of biochemical and structural characterization of screening hits to ensure they will be of functional relevance in cells.

H314 binds at the very N-terminus of CUL5, the binding site of the adapter complex SOCS2-ELOBC (Supplementary Fig. 2d), consistent with its disruption of the CUL5 interaction with ELOBC in SPR assays. Meanwhile, H313 binds at a novel VHL site that is distinct from and adjacent to the HXC78-binding site, which to our knowledge was the only known VHL ligand-binding site prior to this work (Supplementary Fig. 2d).

The distinct binding modes and Helicons described here represent promising hits for further development of probes and tools for this important E3 family, though experimental validation in cells will ultimately be required to assess how functional the Helicon complexes are. For instance, both the H314-binding site on CUL5 and the H313-binding site on VHL face the Ub-E2 binding sites of the cognate Cullin

CTDs (Supplementary Fig. 2d), providing a potentially ideal configuration for TPD applications, where both the E3 binding site and the 'linkerology' of heterobifunctional molecules is critically important to optimize ternary complex formation and the geometry of the E2- and E3-catalyzed reactions[42]. Defining functionally relevant binding sites on the WWP E3 ligases may also provide future opportunities to develop therapeutics for this target class[28,43].

### Helicons that bind the RING/U-Box E3 family

The fourth and final family of E3 ligases we screened was the RING/U-Box family (Fig. 3a). Within this family, MDM2 and IAPs have been used extensively for TPD[26]. To a limited extent, CHIP/STUB1, which is much more ubiquitously expressed (Supplementary Fig. 1a), has also been used to induce degradation of cancer targets[44,45]. We performed screens of the naive phage library to identify Helicons that bind MDM2 and CHIP (Supplementary Data 1, Fig. 3b). Among the resulting hits, we validated CHIP-binding Helicons H317 and H318 from C80, including by x-ray crystallography to examine binding to their E3 presenters (Fig. 3c, Supplementary Fig. 3a). We also characterized MDM2-binding Helicon H319 from C81, which bound MDM2 with 3.7 μM $K_D$, as determined by SPR. Our co-structure of H317 with CHIP, as well as the previously reported co-structure of MDM2 with ATSP-7041[15]), suggest that these hits bind the E3s in a helical conformation (Supplementary Fig. 3a, b). Surprisingly, while Helicons H317 and H318 belong to the same cluster, we noted several differences in the co-structures with CHIP, reflecting unique conformational changes and binding surfaces revealed by the Helicons. For instance, the side chain of Glu9 in H317 directly interacts with Lys30 of CHIP$^{TPR}$ while Glu10 from H318 does not (Supplementary Fig. 3c).

### Construction of trimerizer libraries for CHIP and MDM2

Having identified a range of Helicon binders across all four main E3 ligase families, we proceeded to build and screen focused libraries for trimerizer discovery. As a proof of concept, we selected hit clusters for the RING/U-Box family E3s CHIP and MDM2, including those exemplified by H319 for MDM2 and H317 and H318 for CHIP, and designed focused libraries by fixing the conserved binding residues and diversifying the surface-exposed Helicon residues (Fig. 1a). We also extended the length of the Helicon library to 20 amino acids to provide additional surface area for target binding. To do this, we maintained the two stapling cysteine and two scaffolding alanine residues inherited from the 14-mer Helicon parents[22], and also fixed consensus residues responsible for binding defined by the cluster logos. The remaining residues were randomized either fully (three to four residues at the ends of the Helicons, plus internal residues that showed no amino acid preference in the cluster logos), or partially (residues that showed some preference for two to five amino acids), using

degenerate or semi-degenerate codons, respectively, in the amplification primers (Fig. 1b). Since the options for semi-degenerate codons are limited by the codon table, we selected those that best represent the distribution in amino acids at a given logo position, which occasionally introduces amino acids that are not preferred or tolerated for presenter binding. The focused library primers were synthesized and stapled phage libraries were generated in the same manner as the naive library used for the first-round screens.

To perform trimerizer phage screening, we incubated purified, bead-immobilized target proteins with the phage libraries – in this case, the focused libraries built for discovery of CHIP and MDM2 trimerizers – but also included non-immobilized E3 presenter proteins free in solution in certain wells, as described above (Fig. 1c). As we were specifically interested in identifying Helicon hits that promoted the formation of ternary complexes, we focused on hits that bound the target only in the presence of the E3 presenter, and filtered out those that bound in an E3-independent manner. Screens containing target-free blank beads with free presenter were also included for use as a counter-target. We performed screens to identify such presenter-dependent trimerizers for CHIP and TEAD4, CHIP and PPIA, and MDM2 and β-catenin.

### Trimerizer Helicons that induce the interaction between CHIP and TEAD4 or PPIA

TEAD4 is a member of the transcriptional enhancer factor (TEF) family of transcription factors, and through its interactions with YAP/TAZ, acts as an effector of the Hippo signaling pathway, which is implicated in cell proliferation and migration, organ development, and resistance to specific cancer treatments[46]. We assessed the ability of CHIP-TEAD4 trimerizers hits to promote ternary complex formation using time-resolved fluorescence energy transfer (TR-FRET), SPR (ABA mode), and fluorescence polarization (FP) assays (Fig. 4).

Using TR-FRET, we tested Helicon H321 from C85 and a C-terminally truncated version of H321, H322 (Fig. 4a). We also tested Helicons H323 and H324 from the same cluster and a control peptide, P325, a heterobifunctional chimeric peptide generated by fusing CHIP- and TEAD4-interacting peptides with a connecting linker[47,48] (Supplementary Fig. 4a). Models of PROTAC-mediated ternary complex formation predict that high concentrations can saturate binding to the target, producing ineffective binary complexes that limit ternary complex formation, and result in a bell-shaped dependency on PROTAC concentration – a phenomenon referred to as the "hook effect"[49]. Theoretically, the hook effect can be circumvented by improving the cooperative binding of the PPIs[50]. As expected, we observed a hook effect with the chimeric P325, while H321 - H324 showed a dose-dependent increase in ternary complex formation over the range of concentrations we tested. Crucially, an ABA-format SPR assay

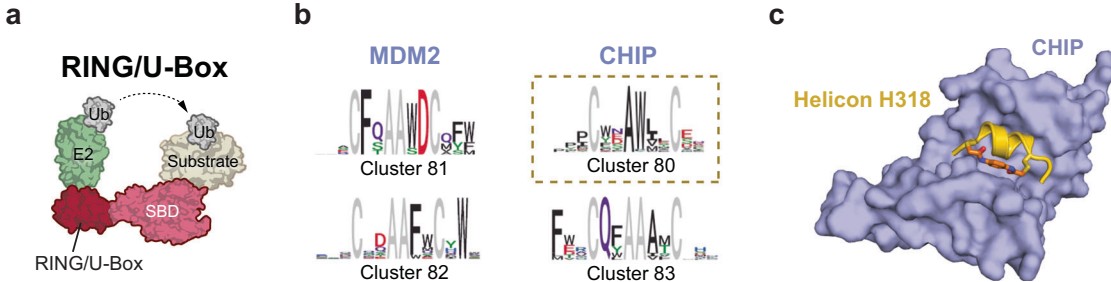

**Fig. 3 | Discovery of Helicon binding sites for members of the RING/U-Box E3 ligase family. a** A prototypical RING/U-Box E3 ligase consisting of a single RING domain and a direct E2-to-substrate catalytic mechanism. The U-Box domain is at the C-terminus and serves as the binding site for the ubiquitin-charged E2 ligase and acts to promote ubiquitin transfer. **b** Representative clusters and their logos

derived from screening a Helicon library for binders to MDM2 (residues 25–109) and CHIP (residues 23–303, CHIP$^{23-303}$). **c** Helicon H318 was crystallized with the TPR domain of CHIP (residues 24–154, CHIP$^{TPR}$) and the co-structure was solved at 1.47 Å resolution (PDB: 8EIO). The electron density map of Helicon H318 is shown in Supplementary Fig. 7.

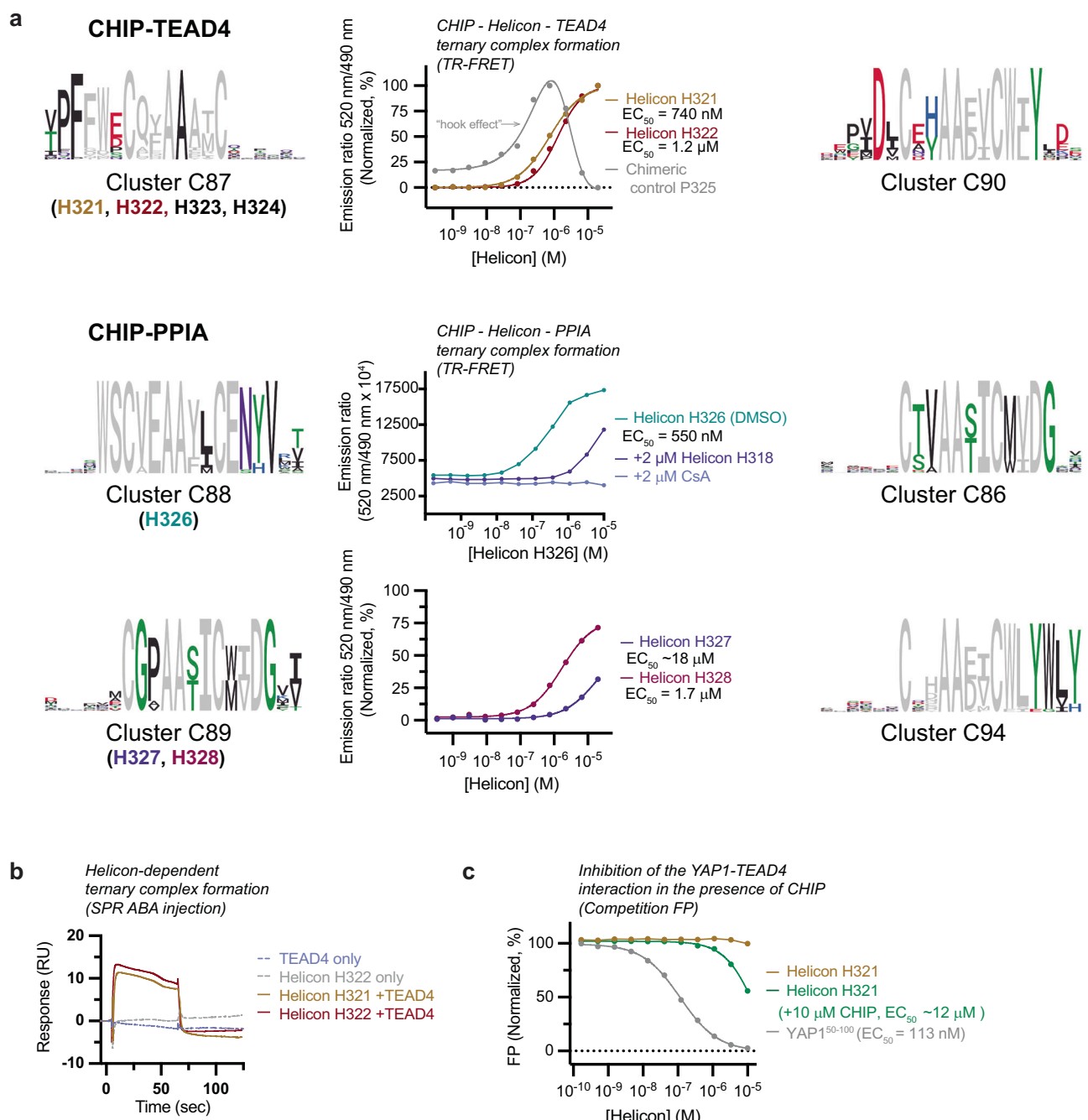

**Fig. 4 | Discovery and characterization of trimerizer Helicons that induce the interaction between CHIP and TEAD4 or PPIA. a** Trimerizer clusters and ternary complex formation for Helicons that bind TEAD4 in a CHIP-dependent manner (CHIP-TEAD4) and those that bind PPIA in a CHIP-dependent manner (CHIP-PPIA). Two clusters are shown for CHIP-TEAD4 and four are shown for CHIP-PPIA. For TR-FRET, Helicons H321 and H322 from C87 were added at increasing concentrations to biotinylated TEAD4 (residues 217–434) and Alexa Fluor 488-labeled CHIP[23–303]. FRET signal indicating CHIP-TEAD4 binding was monitored upon addition of terbium (III)-labeled streptavidin. The chimeric (heterobifunctional) peptide control P325 consists of CHIP- and TEAD4-interacting peptides[47, 48] connected by a linker, and displays the expected "hook" effect[49]. Similarly, CHIP-PPIA ternary complex formation mediated by Helicons H326-H328 was evaluated with TR-FRET between biotinylated full-length PPIA and Alexa Fluor 488-labeled CHIP[23–303]. Ternary complex formation induced by H326 can be competed by CsA and H318. $n = 2$; data are presented as the mean of technical replicates. Where direct binding data were normalized to better visualize the magnitude of the responses, this was done by determining the mean of each data point, setting the smallest mean (background signal) to 0%, and the highest mean (maximum ternary complex for each peptide) to 100%. TR-FRET data that have not been normalized are graphed in Supplementary Figs. 8b, d. **b** SPR assays in ABA mode with immobilized CHIP[23–303] show H321 and H322-dependent binding of TEAD4 to CHIP, $n = 1$. **c** Helicons were added at various concentrations to TEAD4 and a fluorescent YAP peptide (residues 50–100) for Competition FP assays. H321 interfered with the YAP1-TEAD4 interaction only in the presence of CHIP. An unlabeled YAP1 fragment (residues 50–100) was included as a positive control. $n = 2$; data are presented as mean of technical replicates. Competition data were normalized by determining the mean of each data point, setting the smallest mean (maximum competition of the complex by each peptide) to 0% and the largest mean of each data point (minimum competition of the complex by each peptide) to 100%. Competition FP data that have not been normalized are graphed in Supplementary Fig. 8e. Source data are provided as a Source Data file.

demonstrated that CHIP bound to TEAD4 only in the presence of H321 and H322, consistent with cooperative formation of the ternary complex (Fig. 4b).

We also observed that both H321 and H322 could disrupt the interaction between the YAP/TAZ-binding domain of TEAD4 and a fluorescently labeled YAP1 fragment in the presence of CHIP in a competition FP assay (Fig. 4c, Supplementary Fig. 4b). As expected, an unlabeled YAP1 fragment could also disrupt the interaction (Fig. 4c, Supplementary Fig. 4b). Finally, we confirmed that while H321 and H322 do not interact with TEAD4 alone (Fig. 4b), they did show weak binding affinity to CHIP, at a higher $EC_{50}$ than the CHIP-binder H318 characterized in Fig. 3c (Supplementary Fig. 4b). The chimeric P325 also binds to CHIP with a higher $EC_{50}$ than H318 (Supplementary Fig. 4b). These results suggest that besides inducing ternary complex formation only in the presence of CHIP, CHIP-TEAD4 trimerizers act to functionally disrupt the biologically relevant TEAD4-YAP1 interaction.

To test the versatility of CHIP to engage other targets beyond TEAD4, we screened for trimerizers between CHIP and another model protein, PPIA (Cyclophilin A). PPIA is a peptidyl-prolyl cis-trans isomerase (PPIase) that plays a widespread role in the folding of nascent proteins. Screens with the CHIP-based trimerizer library identified several CHIP-dependent PPIA clusters, including C86, C88, C89, and C94 (Fig. 4a). Helicon H326 from C88 could promote ternary complex formation between CHIP and PPIA as shown by TR-FRET, and this was abolished by unlabeled CsA and diminished by the CHIP-binder H318 described in Fig. 3c (Fig. 4a), suggesting that the complex forms via the CsA-binding site of PPIA and the H318-binding site of CHIP. C89 Helicons H327 and H328 similarly acted as trimerizers between the two proteins as shown by TR-FRET (Fig. 4a).

## Trimerizer Helicons that induce the interaction between MDM2 and β-catenin

Having established the ability to discover trimerizers with a first E3 ligase, we turned to a second, MDM2, to assess the generality of our approach. We screened MDM2-based focused libraries against β-catenin, a key component of the canonical Wnt signaling pathway that is often dysregulated in cancer[51]. This screen identified hits belonging to multiple clusters, including C91-C93, that bound β-catenin only in the presence of MDM2 (Fig. 5a). We characterized representative Helicons from each of these three clusters for their ability to act as trimerizers by FP, SPR, and x-ray crystallography.

First, we used a direct FP assay of ternary complex formation to show that H329 and H330 from C91, H332 and H333 from C92, H334 from C93, and an N-terminally truncated version of H329 (H331) could all promote a cooperative interaction between MDM2 and the Armadillo domain of β-catenin with $EC_{50}$ values ranging from 10 to 100 nM (Fig. 5b). Notably, none of the Helicons that we tested could promote ternary complex formation between β-catenin and MDM4, which shares ~55% sequence identity with MDM2 in the p53-binding domain used in these studies, demonstrating the ability of these trimerizer Helicons to bind with selectivity against a related family member (Supplementary Fig. 4c). H330 could also promote the interaction between MDM2 and the full-length β-catenin (Supplementary Fig. 4c). As expected, a control heterobifunctional molecule P335 made by fusing MDM2- and β-catenin-interacting peptides with a connecting linker exhibited a hook effect (Fig. 5b). By competition FP, we found both Helicon 330 and H332 possessed very weak binding to MDM2 alone (Supplementary Fig. 4c). Meanwhile, by SPR, we found that H330, but not H332, could weakly interact with β-catenin in the absence of MDM2 (Supplementary Fig. 4c), while both Helicons could promote the interaction between MDM2 and immobilized β-catenin, but with different affinities (Fig. 5c).

We further confirmed the H330 trimerizer activity in an inverted SPR experiment, where MDM2 was immobilized with β-catenin as the free analyte (Supplementary Fig. 4c). The MDM2/MDM4-binding helical peptide ATSP-7041[15] did not promote ternary complex formation, consistent with the results of the direct FP assay (Fig. 5b, Supplementary Fig. 4c). H330 could compete with ATSP-7041 for binding to MDM2 and with the β-catenin-binder ICAT[52] for binding to β-catenin (Supplementary Fig. 4d). These results suggested that H330 bridges MDM2 and β-catenin via the ATSP-7041- and p53-binding site on MDM2 and the ICAT-binding site on β-catenin.

## Binding kinetics and quantitation of trimerizer activity

To further characterize their activity, we performed SPR (ABA) binding assays for trimerizer Helicons that induce interactions between CHIP and TEAD4, CHIP and PPIA, and MDM2 and β-catenin. First, we assessed the binding affinities ($K_D$) for the CHIP-TEAD4 interactions induced by Helicons H321 and H323, and found that the values (150 and 254 nM, respectively) were consistent with the $EC_{50}$ values determined for ternary complex formation in the TR-FRET assays (Supplementary Figs. 4a and 5a, Fig. 4a). We also calculated the amount of ternary complex formed and found that H321 showed complete ternary complex formation (101%), and H323 showed nearly complete ternary complex formation with CHIP and TEAD4 (65%). Similarly, Helicons H326 and H328 induced complex formation with CHIP and PPIA with $K_D$ values that were consistent with $EC_{50}$ values determined by the TR-FRET assays (Supplementary Fig. 5b, Fig. 4a). Helicon H327, another of the CHIP-PPIA trimerizers, did not show activity in this assay (Supplementary Fig. 5b). In contrast to the SPR results obtained for MDM2 and β-catenin and CHIP and TEAD4, we observed unexpectedly that Helicons H326 and H328 induced ternary complex formation with CHIP and PPIA at only 3% and 11% of the expected response in this assay. We speculate that this may be due to steric hindrance between the immobilized streptavidin tetramer on the SPR sensor chip and the immobilized PPIA.

We next tested the MDM2-β-catenin trimerizer Helicons H329, H330, H332, H333, and H334 in this SPR assay and found that while binding was detectable, each of the Helicons showed 5- to 10-fold weaker affinities than we had determined using the in-solution FP assays (Supplementary Fig. 5c, Fig. 5b). However, we were able to calculate that all five of these trimerizer Helicons could induce ternary complex formation, ranging between 50% and 80% of the expected response. These results further support the ability of Helicons from these trimerizer screens to induce ternary complex formation and new interactions between CHIP and MDM2 and their targets.

## Structural characterization of MDM2-β-catenin trimerizer Helicons reveals direct interactions between all three molecules of the complex

Finally, we performed x-ray crystallography to characterize the MDM2-β-catenin complexes induced by trimerizers H329 and H330 from C91 and H332 and H333 from C92 (Fig. 6a, Supplementary Fig. 6a, b). The calculated electron density maps indicated that residues 6–21 of H330 and residues 5–21 of H332 were well-resolved. Of note, the structures of MDM2-H332-β-catenin and MDM2-H333-β-catenin were solved in different space groups and with distinct crystal packings, but resolved a similar MDM2-β-catenin interaction, excluding potential artifacts arising from crystallography. Gratifyingly, all four structures confirmed Helicon-mediated ternary complex formation, with the Helicons bridging the p53-binding site of MDM2 and the C-terminal ICAT-binding site of β-catenin. The Helicons from the two clusters engage different subsites and residues on the surface of the proteins, revealing distinct structural solutions to cooperative ternary complex formation.

Using the PDBePISA explorer[53] to define the macromolecular interfaces between Helicons H330 and H332 and MDM2 or β-catenin, and between MDM2 and β-catenin, we observed that both Helicons induced similarly sized MDM2-β-catenin interfaces, though H330 did

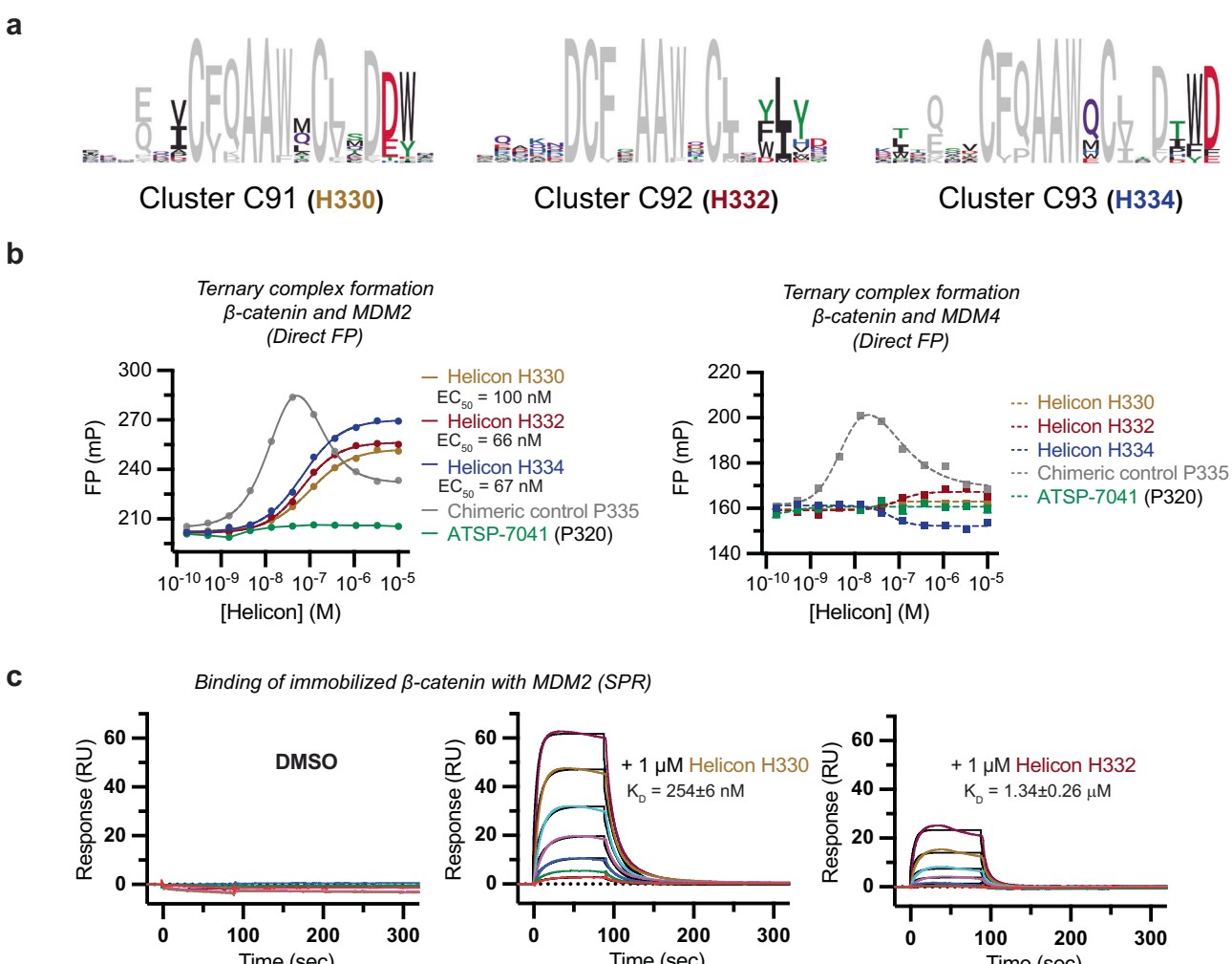

**Fig. 5 | Discovery and characterization of trimerizer Helicons that induce the interaction between MDM2 and β-catenin. a** Three trimerizer clusters that cooperatively induce the interaction between MDM2 and β-catenin. **b** Ternary complex formation was assessed using direct FP, where Helicon and β-catenin are incubated with fluorescent p53-binding domains of MDM2 (residues 17–111) or MDM4 (residues 14–111) over a range of Helicon concentrations. Helicons H330, H332, and H334 from trimerizer clusters C91, C92, and C93, respectively, as well as the chimeric (heterobifunctional) peptide P335 consisting of linked MDM2- and β-catenin-interacting peptides can promote ternary complex formation (left). The chimeric peptide P335 demonstrates the expected hook effect of a hetero-bifunctional molecule. Ternary complex formation was dependent on MDM2 and

did not occur with H330, H332, or H334 when using the MDM2 homolog MDM4, which shares ~55% sequence identity with MDM2 in the p53-binding domain (right). P335 is able to bind to MDM4 to promote a ternary complex, as expected. $n = 2$; data are presented as mean of technical replicates. **c** SPR was performed by immobilizing the Armadillo domain of β-catenin (residues 134–665) and flowing through the p53-binding domain of MDM2 (residues 17–111), in the presence of DMSO vehicle (left), 1 μM H330 (middle), or 1 μM H332 (right). Binding response is seen only when all three components are present. Concentrations of MDM2 from bottom to top in the sensorgram are 2-fold dilutions from 625.0 nM to 9.8 nM. $n = 1$. Source data are provided as a Source Data file.

so with a much more extensive set of interactions with β-catenin than H332 did (Supplementary Table 3, Supplementary Fig. 6b).

In the 2.6 Å ternary structure between MDM2, Helicon H330, and β-catenin, we found that the sidechains of Tyr9, Trp13 and Leu16 of H330 insert deeply into a hydrophobic cleft on the MDM2 surface, reminiscent of the endogenous p53-MDM2 interaction mediated by p53 residues Phe19, Trp23, and Leu26 (ref. 54) (Supplementary Fig. 6b). The interactions between H330 and β-catenin include a series of intramolecular hydrogen bonds, including between Arg582 of β-catenin and Asp19 of H330, between Arg612 of β-catenin and Asp18 of H330, and between His578 of β-catenin and Asp18 of H330 (Fig. 6b). Interestingly, Asp19 of H330 also directly interacts with His96 of MDM2, serving as a bridge for the assembly of the ternary complex. H330 and H332 also use hydrophobic interactions to interact with β-catenin (Supplementary Fig. 6c). Finally, we observed engagement of H330 with the main chain carbonyl group of MDM2 Val109 and the sidechain of β-catenin Lys433 (Fig. 6b, Supplementary Fig. 6c).

Similar to the H330-mediated ternary complex, the 3.9 Å ternary structure between MDM2, Helicon H332, and β-catenin again involved Helicon residues Phe9, Trp13, and Ile16 for interacting with MDM2, but revealed a distinct solution for MDM2-Helicon-binding to β-catenin, with notable differences at the β-catenin-binding interface (Fig. 6c, Supplementary Fig. 6b). Specifically, the H332 β-catenin interaction is driven by the C-terminal hydrophobic tail residues of the Helicon, Trp18, and Ile19, engaging β-catenin residues including Tyr654 (Fig. 6c). We also observed several Helicon-driven hydrogen bonds and salt bridges between MDM2 and β-catenin, including Gln71 and His73 of MDM2 interacting with Glu664 of β-catenin, and His96 of MDM2 interacting with Gln623 of β-catenin. Finally, Phe660 of β-catenin is positioned in a hydrophobic pocket surrounded by His73 of MDM2 and the staple residue of H332 (Fig. 6c). These results are consistent with H330 and H332 belonging to separate trimerizer clusters, with different exposed residues on the helix opposite the MDM2-binding face, and also validate the two-step screening approach, where fixed

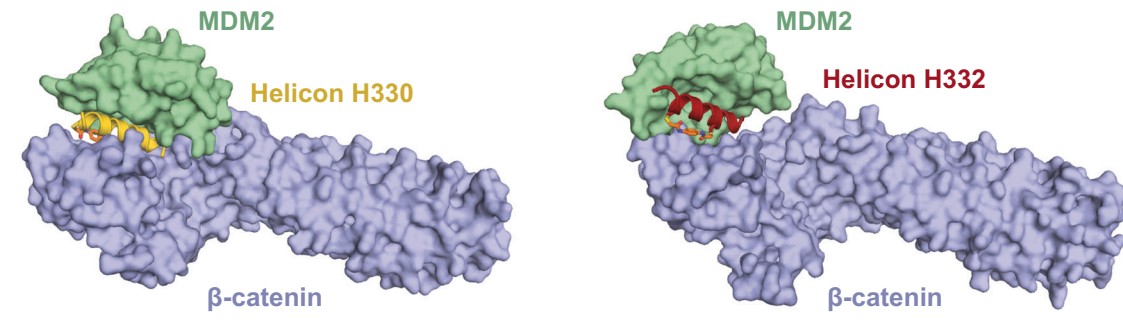

*Co-crystal structures of β-catenin and MDM2 with Helicons*

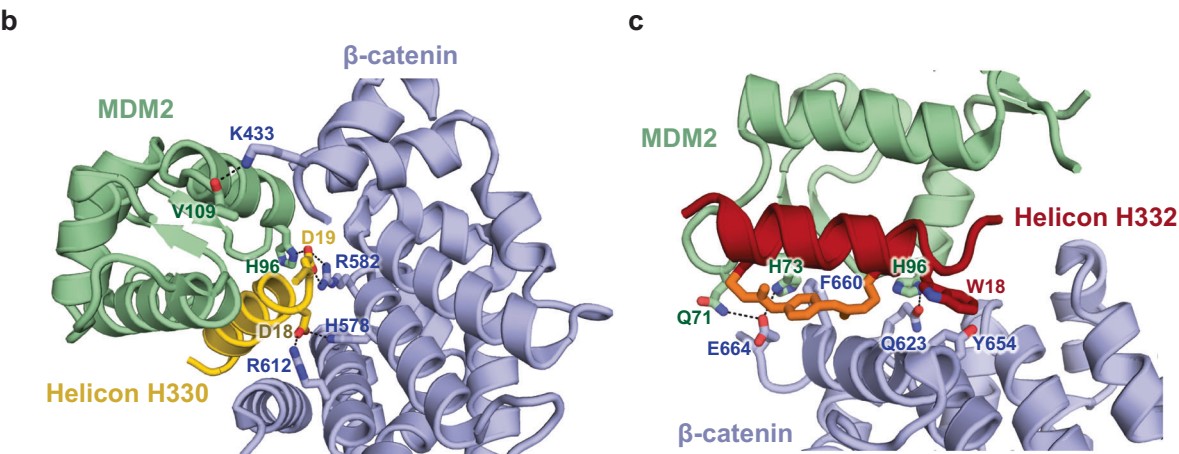

**Fig. 6 | Trimerizer Helicons promote cooperative interactions between MDM2 and β-catenin. a** X-ray co-crystal structures of H330 (PDB: 8EIC) and H332 (PDB: 8EI9) in complex with MDM2 (residues 17–111) and β-catenin (residues 134–665). Below are close-up views of the **b** H330- and **c** H332- mediated ternary complexes

formed with MDM2 and β-catenin, revealing specific residue-level molecular recognition events that drive cooperative formation of complexes. Electron density maps of these Helicons are shown in Supplementary Fig. 6b.

Helicon residues bind the E3 and the opposite Helion face binds to the therapeutic target.

In total, these structures reveal the interactions involved in trimerizer-induced molecular recognition events that promote cooperative binding between the E3 and target proteins. As exemplified by the four trimerizer Helicons that drive formation of ternary complexes with MDM2 and β-catenin, a small number of specific residues within all three components work in a concerted fashion to stabilize distinct PPI structures.

## Discussion

Just over a decade ago, the E3 ligase family was considered largely undruggable by traditional small molecules[55,56], and even today only a handful of members of this large family have been targeted. Using our high-throughput screening platform, we identified dozens of Helicons that bind, and in some cases modulate, diverse E3s across all four major families, thereby adding to the targeted degradation toolbox that currently includes the E3s VHL, CRBN, MDM2, and IAP that are used in a majority of TPD applications. Of note, this includes E3s with distinct tissue distributions (Supplementary Fig. 1a), which may offer the potential to tune cellular selectivity and therefore improve toxicity profiles in specific therapeutic contexts.

An important component of the target-binding specificity of molecular glue-like molecules is the cooperativity in binding that results from specific glue-induced molecular recognition. Several structural and thermodynamic studies have shown a strong correlation between the cooperative formation of ternary complexes and the

efficiency with which they can act as degraders. For instance, dependent on the solvent-exposed moieties of the molecular glues thalidomide and its derivatives pomalidomide and lenalidomide, the surface of the E3 Cereblon (CRBN) can be allosterically modified by these glues to shift the enzyme's substrate selectivity[57–60]. These studies and others[7] have revealed a tight correlation between the PPIs, cooperativity, and stable ternary complex formation that guide selectivity, and also highlight the difficulty in engineering cooperativity in molecular glues, particularly as stabilizing PPIs also requires high-affinity binding. Indeed, the activity of many of the classical molecular glues such as thalidomide and rapamycin were discovered serendipitously[61].

This challenge motivated us to develop a straightforward, rapid screening platform to convert E3-binding Helicons into "trimerizers" that reprogram the E3 surface to cooperatively bind a new target protein for which it previously had no affinity. This method does not rely on rational design or previously known binding ligands, and does not require any preexisting structural information. Rather, it involves a straightforward procedure of performing a single-step naive screen for one protein, followed by the design of a second library based only the data obtained in the naive screen, followed lastly by a second single-step screen with clear hit selection criteria to identify cooperative trimerizers. We note that, in the context of DNA-encoded libraries, a conceptually similar screening approach has been used to identify small-molecule degraders of the classic TPD substrate BRD4. In that work, a single-round of screening of a DNA-encoded library in the presence of both presenter VHL-ELOBC and target protein led to the discovery of compounds that cooperatively bridged the two

proteins[62]. Another recent study developed a shotgun approach using DNA-encoded library technology to screen in parallel for ternary complex formation and cooperative binding to discover BRD4-degrading PROTACs[63]. Having identified Helicons that reprogram E3 surfaces to recognize new targets using our platform, the next step towards developing tools that could be used for induced-proximity applications such as TPD will be to focus on optimizing their cellular penetration and assessing their ability to induce ubiquitylation and neo-substrate degradation in cells and in vivo.

The importance of adopting a two-step screening approach with the creation of a focused library is highlighted by the trimerizer cluster logos we identified, many of which include eight or more positions that are limited to just one or two possible amino acids. As the number of positions that have a narrow tolerance for amino acid identity increases, the expected number of members present in a library of given size will decrease, reducing the likelihood of identifying hits. For example, clusters C86 and C88 contain so many fixed amino acids that that even a single sequence matching their logo might not appear in a library of diversity $10^8$. Indeed, our efforts to directly discover tri-merizers using fully naive screens have met with limited success to date.

We expect that the general approach towards discovering tri-merizers reported here will be amenable to a wide range of targets and presenters beyond E3 ligases, because focused libraries can be designed and built for any protein that can be bound by Helicons, and Helicons can be readily discovered for a wide variety of protein classes. Importantly, because the method reported here relies on commonly available technologies – commercial phage display kits and primer synthesis, standard molecular cloning and protein biochemistry methods, and next-generation amplicon sequencing that can be provided by most sequencing facilities – we are hopeful that it proves broadly accessible to researchers interested in studying and modulating PPIs.

## Methods

### Recombinant protein expression
Unless otherwise stated, all protein constructs correspond to human protein sequences.

**WWP1$^{WW-HECT}$, WWP1$^{HECT}$ and WWP2$^{HECT}$**. The expression and purification of human WWP1 (Uniprot ID: Q9H0M0) and WWP2 (Uniprot ID: O00308) fragments were adapted from previous work[29]. Briefly, WWP1$^{WW-HECT}$ (residues 379–922), WWP1$^{HECT}$ (residues 546–917) and WWP2$^{HECT}$ (residues 492–865) were individually cloned into pET-based expression vectors (Novagen) to generate the final constructs GST-TEV-WWP1$^{379–922}$-yBBr, His-TEV-WWP1$^{546–917}$-yBBr, and GST-TEV-WWP2$^{492–865}$-yBBr, respectively, for phage screening and SPR analysis; and His-Thrombin-WWP1$^{546–917}$, His-3C-WWP2$^{492–865}$ for ELISA and crystallography. Recombinant proteins were expressed in *Escherichia coli* BL21 (DE3) (New England Biolabs). After induction at 16 °C for 16 h with 1 mM isopropyl β-D-1-thiogalactopyranoside (IPTG), the cells were harvested and resuspended in buffer containing 50 mM Tris pH 8.0, 500 mM NaCl, 10% glycerol, and 1 mM phenylmethylsulfonyl fluoride (PMSF). For purification, the pellet was lysed with a tip sonicator, and centrifuged at 22,000 × *g* for 30 min at 4 °C. The supernatant was purified with Pierce™ Glutathione Agarose or Ni-NTA resin (Qiagen), eluting with 50 mM Tris pH 8.0, 500 mM NaCl, 1 mM tris(2-carboxyethyl) phosphine (TCEP), 10% glycerol, with 10 mM reduced glutathione (GSH) or 250 mM imidazole. Eluted proteins were pooled, concentrated, and cleaved by adding the corresponding protease at a protease to protein ratio of 1:10 and incubated overnight at 4 °C. yBBr-tag-containing proteins were biotinylated via the yBBr reaction according to standard procedures[64]. Final proteins were loaded onto a Superdex™ 75 Increase 10/300 GL (Cytiva) size exclusion chromatography (SEC) column and eluted in 50 mM Tris, pH 8.0, 200 mM NaCl, 10% glycerol, 1 mM DTT, and 1 mM EDTA. Fractions containing target protein were collected and pooled. GST contaminants were removed with an additional GST-purification step with target protein collected in the flow-through. Final protein fractions were concentrated and stored at −80 °C.

The yBBr reaction was carried out as previously described[64] with 100 μM target protein tagged with ybbR13 (DSLEFIASKLA), incubated with 150 μM CoA-PEG11-biotin, 5 μM Sfp, and 10 mM MgCl$_2$ in protein storage buffer at room temperature for 1 h. Excess CoA-conjugates and Sfp enzymes were removed by follow-up SEC.

**N-terminal domains of CUL1, CUL2, CUL4B, and CUL5**. For proteins used in phage screening and SPR: the N-terminal domains of CUL1 (Uniprot ID: Q13616, residues 15–410, V367R/L371D), CUL2 (Uniprot ID Q13617, residues 8–384, V340R/L344D), CUL4B (Uniprot ID: Q13620, residues 206–557, V516R/L520D), and CUL5 (Uniprot ID: Q93034, residues 1–386, V341R/L345D) with N-terminal GST-TEV tags and C-terminal AVI tags were cloned into a pET-derived expression vector (Novagen). For proteins used in crystallography: the N-terminal domains of CUL5 (residues 8–384, V340R/L344D) or CUL4B (residues 206–557, V516R/L520D) with N-terminal GST-TEV tags were cloned into pET21b. Proteins were recombinantly expressed in *E. coli* BL21 CodonPlus cells (Agilent). The cells were induced at OD = 0.6 with 1 mM IPTG for 4 h at 37 °C, then harvested and resuspended in buffer, 20 mM HEPES pH 7.5, 300 mM NaCl, 10% glycerol, and 1 mM PMSF. For purification, the pellet was lysed with a tip sonicator, toggling between 3 s on and 3 s off for 20 min, and pelleted at 22,000 × *g* for 30 min. at 4 °C. The supernatant was purified using Pierce™ Glutathione Agarose resin eluting with 20 mM HEPES pH 7.5, 300 mM NaCl, 10% glycerol, and 10 mM GSH. Eluted proteins were pooled, concentrated, and cleaved by adding TEV protease at a ratio of 1:10 protease to protein and incubated overnight at 4 °C. TEV-cleaved proteins were biotinylated with the published AviTag™ technology[65]. Briefly, purified target proteins were incubated with BirA biotin ligase with a 20:1 molar ratio, in a reaction buffer containing 50 μM biotin, 40 μM ATP and 10 mM MgCl$_2$ at 4 °C for 16 h proteins were loaded onto a Superdex™ 75 Increase 10/300 GL (Cytiva) SEC column and eluted in 20 mM Tris pH 7.4, 200 mM NaCl, 2 mM DTT, and 5% glycerol. Fractions containing target protein were collected and pooled. GST contaminants were removed with an additional GST-purification step with target protein collected in the flow-through. Final protein fractions were concentrated and stored at −80 °C.

**VHL, SOCS2, and FBWX7**. For protein used in phage display screens and SPR: FBXW7 (Uniprot ID: Q969H0, residues 263–706) with an N-terminal GST-TEV tag and C-terminal AVI tag were co-expressed with full-length SKP1 (Uniprot ID: P63208, residues 1–163) in the pETDuet-1 plasmid (Novagen). SOCS2 (Uniprot ID: O14508, residues 32–198) or VHL (Uniprot ID: P40337, residues 54–213) with an N-terminal 6xHis-TEV tag cloned into pET21b and co-expressed with C-term AVI-tagged ELOB (Uniprot ID: Q15370, residues 1–104) and ELOC (Uniprot ID: Q15369, residues 17–112) cloned in pCDFDuet-1. For protein used in competition SPR (ABA mode) and x-ray crystallography: SOCS2 (residues 32–198) or VHL (residues 54–213) with an N-terminal 6xHis-TEV tag cloned in pET21b was co-expressed with full-length ELOB (residues 1–118) and ELOC (residues 17–112) cloned in pCDFDuet-1. Recombinant proteins were expressed in *E. coli* BL21 (DE3) host cells and purified and biotinylated as for the WWP and Cullin proteins above.

**MDM2 and MDM4**. For protein used in the phage display screens and SPR: the p53-binding domain of MDM2 (Uniprot ID: Q00987, residues 25–109) with an N-terminal 6xHis-yBBr-TEV tag was recombinantly expressed in *E. coli* BL21 CodonPlus cells (Agilent) from pET-derived expression vectors (Novagen). The cells were induced at OD$_{600}$ = 0.6 with 1 mM IPTG for 4 h at 37 °C, then harvested and resuspended in

25 mM Tris-HCl pH 7.5, 300 mM NaCl, 10% glycerol, 1 mM PMSF. For purification, the pellet was lysed with a tip sonicator, and pelleted at 22,000 × $g$ for 30 min at 4 °C. The pellets were washed three times with 20 mM Tris-HCl pH 8.0, 150 mM NaCl, 1 M urea, 1.0% Triton X-100, and dissolved in 20 mM Tris-HCl pH 8.0, 150 mM NaCl, 8 M urea, and 2 mM β-mercaptoethanol (β-me). The supernatant was purified using a Ni-NTA resin (Qiagen), and eluted with 20 mM Tris-HCl pH 8.0, 150 mM NaCl, 8 M urea, 2 mM β-me, and 250 mM imidazole. Protein elutes were diluted to ~0.1 mg/mL and dialyzed into buffers containing 10 mM Tris-HCl pH 8.0, 150 mM NaCl, 2 mM ß-me, with 4, 2, 1, or 0 M urea, at 4 °C for 8 h for each urea gradient. Urea-free proteins were concentrated with Amicon spin filters (Millipore Sigma) to ~1 mg/mL and biotinylated via the yBBr reaction according to standard procedures[64], as above. Biotinylated proteins were pooled, concentrated, and loaded onto a Superdex™ 75 Increase 10/300 GL (Cytiva) SEC column and eluted in 20 mM HEPES pH 7.0, 200 mM NaCl, 5% glycerol, 0.5 mM TCEP. Fractions containing pure protein were collected, pooled, concentrated to ~1 mg/mL and stored at −80 °C.

For protein used in crystallography and other biochemical assays: p53-binding domains of MDM2 (residues 17–111, with C17S substitution; MDM2[17–111]) with an N-terminal 6xHis-TEV tag, and MDM4 (Uniprot ID: O15151, residues 14–111, with C17S substitution; MDM4[14–111]) with an N-terminal 6xHis-yBBr-3C tag were recombinantly expressed in *E. coli* BL21 (DE3) cells (Agilent) from pET-derived expression vectors (Novagen). The cells were induced at $OD_{600} = 0.6$ with 0.15 mM IPTG for 16 h at 37 °C, then harvested and resuspended in 50 mM Tris, pH 8.0, 200 mM NaCl, 10% glycerol, 1 mM TCEP, and 20 mM imidazole. For purification, the pellet was lysed with a tip sonicator, toggling between 3 s on and 3 s off for 20 min, and then centrifuged at 22,000 × $g$ for 30 min at 4 °C. The supernatant was purified using a Ni-NTA resin (Qiagen), and eluted with 50 mM Tris, pH 8.0, 200 mM NaCl, 10% glycerol, 1 mM TCEP, and and cleaved by adding protease (TEV or PreScission protease) at a protease to protein ratio of 1:10 and incubated overnight at 4 °C. Cleaved proteins were loaded onto a Superdex™ 75 Increase 10/300 GL (Cytiva) SEC column and eluted in 50 mM Tris, pH 8.0, 200 mM NaCl, 10% glycerol, and 1 mM TCEP. Fractions containing pure protein were collected, pooled, concentrated to ~8 mg/mL and stored at −80 °C.

For protein labeling: Purified tag-free MDM2[17–111] and MDM4[14–111] were loaded onto a Superdex™ 75 Increase 10/300 GL (Cytiva) SEC column and eluted in 25 mM HEPES pH 7.5 and 250 mM NaCl, with pooled protein factions with the concentration at 300 µM. The protein was then mixed with Alexa Fluor™ 488 NHS Ester (Thermo Scientific) prepared as 100 mM stock, with a final protein to NHS ratio of 1:0.8. The reaction was carried out at room temperature and quenched with 50 mM hydroxylamine before the final SEC purification in 25 mM HEPES pH 7.5, 250 mM NaCl and 1 mM TCEP buffer. Fractions containing Alexa488-labeled protein were collected, pooled, and stored at −80 °C.

**CHIP.** For protein used in the phage display screens and SPR: N-term truncated CHIP (also known as STUB1, Uniprot ID: Q9UNE7, residues 23–303; CHIP[23–303]) or the TPR domain of CHIP (residues 23–154; CHIP[23–154]) with an N-terminal 6xHis-yBBr-TEV tag was recombinantly expressed in *E. coli* BL21 CodonPlus cells (Agilent) from pET-derived expression vectors (Novagen). The cells were induced at $OD_{600} = 0.6$ with 1 mM IPTG for 4 h at 37 °C, then harvested and resuspended in 50 mM Tris-HCl pH 8.0, 500 mM NaCl, 10 mM imidazole, 10% glycerol, and 10 mM β-me. For purification, the pellet was lysed with a tip sonicator and centrifuged at 22,000 × $g$ for 30 min at 4 °C. The supernatant was purified with Ni-NTA resin (Qiagen), eluted with 50 mM Tris-HCl pH 8.0, 500 mM NaCl, 10% glycerol, 10 mM β-me and 250 mM imidazole, and biotinylated via the yBBr reaction according to standard procedures as above. Biotinylated proteins were pooled, concentrated, and loaded onto a Superdex™ 75 Increase 10/300 GL

(Cytiva) SEC column, and eluted in 20 mM HEPES pH 7.0, 150 mM NaCl, 10% glycerol, 2 mM DTT. Fractions containing pure protein were collected, pooled, concentrated to ~1.2 mg/mL and stored at −80 °C.

For proteins used in crystallography and other biochemical assays: CHIP TPR domain (CHIP[21–154] or CHIP[23–154]), or N-term-truncated CHIP (CHIP[23–303]) each with an N-terminal 6xHis-TEV tag, were recombinantly expressed in *E. coli* BL21 CodonPlus cells (Agilent) from a pET21b-derived expression vector (Novagen). The cells were induced at $OD_{600} = 0.6$ with 1 mM IPTG for 4 h at 37 °C or 16 h at 16 °C, then harvested and resuspended in 50 mM Tris-HCl pH 8.0, 500 mM NaCl, 10 mM imidazole, 10% glycerol, and 10 mM ß-me. For purification, the pellet was lysed with a tip sonicator, toggling between 3 s on and 3 s off for 20 min, centrifuged at 22,000 × $g$ for 30 min at 4 °C. The supernatant was purified with Ni-NTA resin (Qiagen), eluting with 250 mM imidazole. Eluted proteins were pooled, concentrated, and the TEV tag cleaved off by adding TEV protease at a protease to protein ratio of 1:10 and incubated overnight at 4 °C. Cleaved/untagged proteins were loaded onto a Superdex™ 75 Increase 10/300 GL (Cytiva) SEC column, and eluted in 50 mM HEPES, pH 8.0, 150 mM NaCl, 10% glycerol, 2 mM DTT. Fractions containing pure protein were collected, pooled, concentrated to ~30 mg/mL and stored at −80 °C.

For Alexa488 labeling: Purified tag-free CHIP[23–303] was labeled as MDM2[17–111] above, with the final SEC purification in buffer: 20 mM Tris pH 7.5, 250 NaCl, and 1 mM DTT. Fractions containing Alexa488-labeled protein were collected, pooled, concentrated to 0.6 mg/mL, and stored at −80 °C.

**PPIA.** Full length PPIA (Uniprot ID: P62937, residues 1–165) with an N-terminal 6xHis-yBBR-TEV tag was recombinantly expressed in *E. coli* BL21 (DE3) CodonPlus RIPL cells (Agilent) from pET-derived expression vectors (Novagen). The cells were induced at $OD_{600} = 0.6$ with 0.15 mM IPTG for 16 h at 16 °C, then harvested and resuspended in PBS pH 7.4 with 1 mM PMSF. For purification, cell pellets were lysed with a tip sonicator and centrifuged at 22,000 × $g$ for 30 min at 4 °C. The resulting supernatant was collected then centrifuged again at 22,000 × $g$ for 30 min at 4 °C. The supernatant was purified with Ni-NTA resin (Qiagen) and eluted with 250 mM imidazole. TEV was cleaved from the recombinant proteins by adding TEV protease at a protease to protein ratio of 1:10 and incubation for 4 h at 4 °C. Protein was then concentrated and diluted into 20 mM HEPES pH 7.0, 5% glycerol, and centrifuged at 22,000 × $g$ for 10 min at 4 °C. The supernatant was loaded onto an SP HP (Cytiva) column pre-equilibrated with 20 mM HEPES pH 7.0, 5% glycerol. Purified protein was eluted with a gradient from 0 mM to 1 M NaCl. Protein fractions were pooled, concentrated then centrifuged at 22,000 × $g$ for 10 min at 4 °C. The supernatant was collected and loaded onto a HiLoad 16/600 Superdex™ 200 pg (Cytiva) SEC column pre-equilibrated with PBS pH 7.4. Purified proteins were eluted isocratically in PBS pH 7.4. Protein fractions were collected, concentrated, aliquoted and frozen.

**TEAD4.** YAP/TAZ-binding domain of TEAD4 (Uniprot ID: Q15561, residues 217–434) with an N-terminal 6xHis-yBBr-TEV tag was recombinantly expressed in *E. coli* BL21-CodonPlus cells (DE3) (Agilent) from pET-derived expression vectors (Novagen). The cells were induced at $OD_{600} = 0.6$ with 0.15 mM IPTG for 16 h at 16 °C, then harvested and resuspended in 50 mM Tris pH 7.4, 200 mM NaCl, 5% glycerol, 1 mM TCEP, 1 mM PMSF. For purification, the pellet was lysed with a tip sonicator, pelleted at 22,000 × $g$ for 30 min at 4 °C, then the supernatant was purified with an Ni-NTA column (Cytiva), eluting with 250 mM imidazole. Protein-containing fractions were pooled, concentrated, and loaded onto a Superdex™ 75 Increase 10/300 GL (Cytiva) SEC column. Purified proteins were eluted isocratically in 50 mM Tris pH 7.4, 200 mM NaCl, 5% glycerol, 1 mM TCEP, and fractions containing pure protein were collected, pooled and frozen.

**β-catenin (CTNNB1).** β-catenin protein (encoded by *CTNNB1*, Uniprot ID: P35222) Armadillo domain (residues 134–665) with a N-terminal 6xHis-yBBr-TEV tag was recombinantly expressed in *E. coli* BL21 (DE3) pLysS cells (Thermo Fisher) from pET28a vectors (Novagen). The cells were induced at $OD_{600} = 0.6$ with 0.15 mM IPTG for 16 h at 16 °C, then harvested and resuspended in 25 mM Tris pH 8.0, 200 mM NaCl, 10% glycerol, 20 mM imidazole, 1 mM PMSF. For purification, the pellet was lysed with a tip sonicator, centrifuged at $22,000 \times g$ for 30 min at 4 °C, then the supernatant was purified with HisTrap HP columns (Cytiva), eluting with 250 mM imidazole. For crystallography and selected biochemical assays, protein was TEV-cleaved by adding TEV protease at a protease to protein ratio of 1:10 and incubated overnight at 4 °C. For phage display screening and SPR analysis, protein was biotinylated via the yBBr reaction according to standard procedures. All proteins were concentrated using Amicon spin filters (Millipore Sigma) then diluted into 25 mM Tris, pH 8.8, 1 mM DTT, 10% glycerol and loaded onto a Q HP (Cytiva) column. Proteins were eluted with a gradient from 50 mM to 600 mM NaCl. Protein-containing fractions were pooled, concentrated and loaded onto a Superdex™ 75 Increase 10/300 GL (Cytiva) SEC column. Purified proteins were eluted isocratically in 25 mM Tris-HCl, pH 8.8, 10% glycerol, 300 mM NaCl, and fractions containing pure protein were collected and pooled.

Full-length β-catenin protein (CTNNB1, residues 1-781) with N-terminal 6xHis-thrombin-T7-TEV tag was recombinantly expressed in *E. coli* BL21 (DE3) pLysS cells (Thermo Fisher) from pET28a vectors (Novagen). *E. coli* cells were induced at $OD_{600} = 0.6$ with 0.15 mM IPTG for 20 h at 16 °C, shaking at 180 rpm, then harvested and resuspended in 20 mM HEPES, pH 7.5, 300 mM NaCl, 10% glycerol. For purification, the pellet was lysed with a sonicator, pelleted at 130,000 rpm for 30 min at 4 °C, then the supernatant was purified with a Ni-NTA column (Cytiva), eluting with 250 mM imidazole. Protein was then diluted into 20 mM HEPES, pH 7.5, 10% glycerol, 1 mM DTT and loaded onto a Q HP column (Cytiva), and was eluted with a NaCl gradient of concentration 0–1.0 M. Protein-containing fractions were pooled, concentrated, and loaded onto a Superdex™ 75 Increase 10/300 GL (Cytiva) SEC column. Purified protein was eluted isocratically in 20 mM HEPES, pH 7.5, 300 mM NaCl, 10% glycerol and 1 mM DTT, and fractions containing pure protein were collected and pooled.

**ICAT (CTNNBIP1).** Full-length ICAT (CTNNBIP1, Uniprot ID: Q9NSA3, residues 1–81) with an N-terminal GST-TEV tag was recombinantly expressed in *E. coli* BL21-CodonPlus cells (DE3) (Agilent) from pET-derived expression vectors (Novagen). The cells were induced at $OD_{600} = 0.6$ with 0.15 mM IPTG for 16 h at 16 °C, then harvested and resuspended in 20 mM Tris pH 7.4, 200 mM NaCl, 10% glycerol, 0.5 mM TCEP, 1 mM PMSF. For purification, the pellet was lysed with a tip sonicator, pelleted at $22,000 \times g$ for 30 min at 4 °C, then the supernatant was purified with a GST column (Cytiva), eluting with 10 mM glutathione. Protein-containing fractions were pooled, concentrated, and loaded onto a Superdex™ 75 Increase 10/300 GL (Cytiva) SEC column. Purified proteins were eluted isocratically in 20 mM Tris pH 7.4, 200 mM NaCl, 10% glycerol, 0.5 mM TCEP and fractions containing pure protein were collected, pooled and frozen.

### Trimerizer phage library construction (primers, protocol, crosslinking, and DNA sequencing)

**Naive phage library.** The naive phage-displayed Helicon libraries were constructed using previously described methods[22]. Briefly, the Peptide Display Cloning System kit from New England Biolabs was used to construct M13KE-based libraries (New England Biolabs, Ipswich, MA). Library oligonucleotides were chemically synthesized using a mix of trimer phosphoramides (Glen Research, Sterling, VA) without codons encoding cysteine, lysine, proline, or glycine, then annealed, extended, and ligated into a digested M13KE vector. All DNA products were

purified using Monarch PCR and DNA cleanup kit (New England Biolabs, Ipswich, MA). The resulting library-containing phage vector was transformed into *E. coli* strain ER2738 (Lucigen, Middleton, WI) by electroporation and amplified by adding the post-rescue electroporated cells to a 500 mL *E. coli* culture at early-log phase ($OD_{600} = 0.01$). Phage propagation, purification, and stapling were conducted as described previously[22].

**Trimerizer library.** Following identification of Helicon clusters specific for a presenter protein of interest based on screening the naive library, trimerizer library oligonucleotides were designed. Presenter-specific clusters of various sizes were used, ranging in size from 10-mer to 20-mer. As an illustrative example of the design of a trimerizer library, a presenter-specific 20-mer cluster, $X_1X_2 \, X_3X_4W_5E_6C_7X_8E_9A_{10}A_{11}(F/I/L/M)_{12}X_{13}C_{14}X_{15}(F/Y)_{16}(F/Y)_{17}X_{18}X_{19}X_{20}$, will be used. Briefly, codons of conserved or semi-conserved residues responsible for binding with a presenter protein are fixed or partially randomized in the primer to bias the library for retained affinity towards the chosen presenter protein (Fig. 1b). The resulting primer (PR21) was generated as follows: for partial randomization (in parentheses), semi-degenerate codons are used to sample a subset of residues conserved within that position. In this example, position 12 is represented by four potential residues (F/L/I/M) from the identified presenter-specific cluster, so the semi-degenerate codon, WTK is used to code for phenylalanine (TTT), leucine (TTG), methionine (ATG), and isoleucine (ATT) – W represents A or T, and K represents G or T. For positions 16 and 17 within this example, the semi-degenerate codon, TWT, is used to code for both tyrosine (TAT) and phenylalanine (TTT) to represent all residues observed within the identified presenter-specific cluster. All other residues within the displayed Helicon with no apparent preference for presenter binding (represented as Xs within the cluster) are randomized to any other amino acid except cysteine, lysine, proline and glycine. Trimerizer libraries are built using multiple oligonucleotides from various designs based on the presenter specific binding sequences (Supplementary Table 2). Library oligonucleotides are chemically synthesized using a mix of trimer phosphoramides (Glen Research, Sterling, VA) lacking codons for cysteine, lysine, proline, and glycine, annealed, extended, and ligated into a digested with *KpnI* and *EagI* restriction enzymes M13KE vector. The example PR21 oligonucleotide insert coding strand sequence is: 5′-CATGCCCG GGTACCTTTCTATTCTCACTCTGCGCCGXXXXTGGGAATGTXGAAGC AGCAWTKXTGTXTWTTWTXXXGGTGGTTCTGGCGCAGGTCGTGGTT C-3′, where X represents a single trimer phosphoramide incorporation, flanked by a *KpnI* restriction site. The antisense strand complements the 3′ end of the sense strand to allow Klenow extension, 5′-CATGTTTCGGCCGAACCACGACCTGCGCCAGAACCAC-3′. The antisense strand possesses an *EagI* restriction site. Construction of the trimerizer library follows the previously described protocol for construction of the naive library[22].

### Phage library screening

**Naive library presenter screening.** To conduct phage library screening, we followed our previously described procedure[22]. Briefly, Helicon-displayed phage libraries were incubated with streptavidin-coated magnetic beads (Dynabeads MyOne Straptavidin T1, Thermo-Fisher Scientific, Waltham, MA) for 1 h at room temperature in a buffer of 1X TBS, 1 mM $MgCl_2$, 1% (w/v) BSA, 0.1% Tween-20, 0.02% (w/v) sodium azide, 5% (w/v) nonfat milk to deplete the library of bead-binding phage particles. For each screening condition, 100 µL of 2 µM biotinylated protein was captured with streptavidin beads that had been previously blocked with 1% BSA, 0.1% Tween-20, 2% glycerol in 1X TBS pH 7.4 at room temperature for 15 min, the supernatant was removed using a plate magnet and the beads are resuspended in 50 µL of the blocking buffer. 150 µL of the depleted phage library is added to each well for 200 µL final volume, plates are sealed, and the screening

reactions are incubated at room temperature for 45 min, with rotation to maintain beads in solution. Following binding, beads were washed 5 times with ice-cold washing buffer (1X TBS, 1 mM MgCl2, 1% (w/v) BSA, 0.1% Tween-20, 0.02% (w/v) sodium azide, 2% (w/v) glycerol), beads containing protein-bound phage were collected and directly processed for NGS.

**Trimerizer phage screening.** Trimerizer phage screening is performed as for the naive library screening described above, with the key practical difference being that the Trimerizer library is incubated with a presenter protein after removal of the bead-binding phage library members and prior to the incubation with biotinylated proteins bound to streptavidin magnetic beads. To identify presenter-dependent phage-displayed Helicon members, target proteins were screened with the phage library in both the presence and absence of a presenter protein. Prior to addition of the bead-bound targets, the phage library was split into two portions, and the presenter protein was added to one portion to a final concentration of 10 µM. 150 µL of the phage library without presenter protein was then added to a well containing 50 uL of the highest concentration of the target protein, and also to a blank (beads-only) well, both for a final volume of 200 µL. To the remaining wells, 150 µL of the phage library mixed with 10 µM presenter protein was added to wells containing 50 µL of the target of interest at a range of concentrations, and also to a blank (beads-only) well. The plate was sealed, and the screening reactions were incubated at room temperature for 45 min, with rotation to maintain beads in solution. The rest of the experiment was performed similar to the procedure described above with one exception – the addition of a presenter protein at a final concentration of 10 µM to the wash buffer (ice-cold 1X TBS, 1 mM MgCl2, 1% (w/v) BSA, 0.1% Tween-20, 0.02% (w/v) sodium azide, 2% (w/v) glycerol).

**Next-generation sequencing.** Next-generation sequencing was performed as described previously[22]. Briefly, phage particles were denatured from magnetic beads at 95 °C for 15 min with an added spike-in sequence (a non-library member) that is used to enable cross-well normalization of sequence reads, followed by a two-step low-cycled PCR to introduce Illumina adapters and 10 bp IDT for Illumina DNA/RNA UD Indexes (Illumina, San Diego, CA) according to Illumina's 16 S Metagenomic Sequencing Library Preparation protocol. The NGS library was sequenced with an Illumina NovaSeq platform using a 2 × 150 bp high-output kit (Illumina, San Diego, CA).

**Hit ID and clustering.** Hit ID and Clustering is performed according to a previously described procedure[22]. Briefly, NGS reads were trimmed for quality (Phred score ≥18) and filtered for sequences that matched the design of the phage library. Counts for each unique sequence were tallied, and then normalized by the counts of the spike-in sequence added to each sample. A metric called Hit Strength was computed for each sequence as the fold-change between the normalized counts in the highest target concentration sample with presenter and the normalized counts in the target (no presenter) samples (averaged across experimental replicates). By using target wells with no presenter as "target blanks", presenter-dependent binding could be identified. This approach eliminates sequences that show binding to target alone, or binding to a free presenter alone. When 0 counts are observed for a sequence in target only "target blank" samples, a count of 0.5 is used to prevent dividing by zero (Supplementary Data 1). Sequences with a hit strength greater than 5 were subjected to hierarchical clustering to identify sequence families[22].

**Helicon synthesis**
The synthesis of the cysteine-stapled Helicons was previously reported[22]. Briefly, linear peptides containing two cysteine residues were synthesized at 100 or 250 µmol scale on Rink Amide resin

(~0.5 mmol/g) using standard Fmoc-based solid phase peptide synthesis workflows. The peptides were globally deprotected and cleaved off-resin, then dissolved in DMSO. The DMSO stock was diluted in a 2:1 solvent mixture of acetonitrile and 50 mM ammonium hydroxide. The pH of the solution was adjusted to ~8.5 using N,N-Diisopropylethylamine (DIPEA). For crosslinking of cysteine residues, ~1.3 equivalents of the alkylating agent, N,N'-(1,4-phenylene)bis(2-bromoacetamide) in DMF were added to the crude peptide solution for two hours at room temperature. The crude helicons were purified by preparatory HPLC, and the purity of the final products were analyzed with analytical UPLC. The R8-S5 stapled peptides, including ATSP-7041 (P320), were synthesized as described previously[15].

**Surface plasmon resonance (SPR) spectroscopy**
**SPR screen of E3-binding helicons.** To confirm Helicon binding to all selected E3 ligases and E3-related proteins, SPR experiments were performed on a Biacore 8 K (Cytiva) instrument at 25 °C in 1x HBS-P+ buffer (Cytiva) with 1% DMSO. A SA Series S sensor chip was docked and pre-conditioned with three injections of 50 mM NaOH/1 M NaCl to remove unbound streptavidin from the surface. Biotinylated proteins, including CUL4B^NTD, CUL5^NTD, WWP1^WW-HECT, WWP1^HECT, WWP2^HECT, VHL-ELOBC, SOCS2-ELOBC, CHIP^23–154, CHIP^23–303, and MDM2^5–109, were each diluted to 5–10 µg/mL in running buffer and immobilized to channels 1 through 8 at 5 µL/min for 50–80 s for a final immobilization level of ~500–2000 RU. Helicons were diluted to 5 µM in running buffer and then serially diluted 2-fold for a total of seven concentrations with one blank (7-point two-fold Helicon dilution series with top concentration = 5 µM and bottom concentration = 78 nM). Compounds were injected over the immobilized and reference surfaces at 30 µL/min for 60 s and then allowed to dissociate for 180 s without surface regeneration ($n = 1–2$). Data were analyzed using Biacore Insight Evaluation Software (Cytiva). Sensorgrams were double referenced, with most of them fitted to a 1:1 steady-state affinity model, with a few fitted with both the steady-state affinity model and the 1:1 binding kinetic model.

**SPR analysis of the trimerizer Helicon-dependent ß-catenin:MDM2 interaction.** To probe the trimerizer Helicon-dependent CTNNB1:MDM2 interaction, SPR experiments were performed on a Biacore S200 (Cytiva) instrument at 25 °C in 1x HBS-P+ buffer (Cytiva) with 1% DMSO. CTNNB1^134–665 was immobilized using the Biotin CAPture Kit, Series S (Cytiva) to ~600–1000 RU. Tag-free MDM2^17–111 was diluted to 625 nM then serially diluted 2-fold for a total of seven concentrations with one blank (7-point two-fold Helicon dilution series with top concentration = 625 nM and bottom concentration = 9.8 nM), in running buffer in the absence or presence of 1 µM trimerizer Helicon. MDM2-binding Helicons were injected over the immobilized and reference surfaces at 30 µL/min for 90 s and then allowed to dissociate for 270 s. The chip surface was regenerated with a 120-second injection of CAP regeneration solution each cycle. Data were analyzed using Biacore Insight Evaluation software (Cytiva). Sensorgrams were double-referenced and evaluated for competition.

**SPR analysis of trimerizer Helicons against ß-catenin.** To understand how the trimerizer Helicons interact with CTNNB1 by themselves, SPR experiments were performed on a Biacore S200 (Cytiva) instrument at 25 °C in 1x HBS-P+ buffer (Cytiva) with 1% DMSO. CTNNB1^134–665 was immobilized using the Biotin CAPture Kit, Series S (Cytiva) to ~600–1000 RU, while Helicons were diluted to 10 µM in running buffer then serially diluted 2-fold for a total of four concentrations with one blank (4-point two-fold Helicon dilution series with top concentration = 10 µM and bottom concentration = 1.25 µM). Data were analyzed using Biacore Insight Evaluation software (Cytiva). Sensorgrams were double-referenced and evaluated for affinity.

**SPR competition, ABA mode: CUL5: SOCS2-ELOBC.** To probe the H314-binding site on CUL5, SPR ABA experiments were performed on a Biacore S200 (Cytiva) instrument at 25 °C in 1x HBS-P+ buffer (Cytiva) with 1% DMSO. Biotinylated CUL5 (residues 1–186) was immobilized using the Biotin CAPture Kit, Series S (Cytiva) to ~150 RU. For each injection, H314 in 10 mM DMSO stock was diluted to 10 μM in SPR running buffer and was injected over the surface for 120 s at 30 μL/min to achieve equilibrium binding. 100 nM tag-free SOCS2-ELOBC was then injected for 60 s at 30 μL/min in the absence or presence of H314 over the surface. The chip surface was regenerated with a 120-second injection of CAP regeneration solution each cycle. Data were analyzed using Biacore Insight Evaluation software (Cytiva). Sensorgrams were double-referenced and evaluated for competition.

**SPR, ABA mode: TEAD4: CHIP.** To confirm the ternary complex formation of TEAD4: CHIP, SPR ABA experiments were performed on a Biacore S200 (Cytiva) instrument at 25 °C in 1x HBS-P+ buffer (Cytiva) with 1% DMSO and 0.05 mM TCEP. Biotinylated CHIP$^{23–303}$ was immobilized using the Biotin CAPture Kit, Series S (Cytiva) to ~500 RU. For each injection, Helicons in 10 mM DMSO stock were diluted to 10 μM in SPR running buffer and were injected over the surface for 120 s at 30 μL/min to achieve equilibrium binding (A). 300 nM tag-free TEAD4$^{217–434}$ protein was then injected for 60 s at 30 μL/min (B) in the absence or presence of Helicons over the surface. The chip surface was regenerated with a 120-second injection of CAP regeneration solution each cycle. Data were analyzed using Biacore Insight Evaluation software (Cytiva). Sensorgrams were double-referenced and evaluated for competition.

For kinetics experiments of the TEAD4:CHIP ternary complex, SPR ABA experiments were performed on a Biacore S200 (Cytiva) instrument at 25 °C in 1x HBS-P+ buffer (Cytiva) with 1% DMSO and 0.5 mM TCEP. Biotinylated TEAD4 was immobilized using the Biotin CAPture Kit, Series S (Cytiva) to ~300–400 RU. For each injection, Helicons in 10 mM DMSO stock were diluted to 10 μM in SPR running buffer and were injected over the surface for 120 s at 30 μL/min to achieve equilibrium binding. 2.5 μM tag-free CHIP protein was serially diluted 1:3 for 8 total concentrations, then injected over the surface for 180 s at 30 μL/min in the presence of 10 μM Helicon. The chip surface was regenerated with a 120-second injection of CAP regeneration solution each cycle. Data were analyzed using Biacore Insight Evaluation software (Cytiva). Sensorgrams were double-referenced and fit to steady state affinity models to determine $K_D$ and $R_{max}$. The percent of Helicon-induced ternary complex was calculated by first taking the molecular weights of each component of the ternary complex to determine the expected response in RU. This value was then divided by the experimental response ($R_{max}$) and multiplied by one hundred to determine the percent of complex formed.

**SPR, ABA mode: MDM2:β-catenin.** To confirm the ternary complex formation of MDM2:β-catenin, SPR ABA experiments were performed on a Biacore S200 (Cytiva) instrument at 25 °C in 1x HBS-P+ buffer (Cytiva) with 1% DMSO and 0.5 mM TCEP. Biotinylated MDM2$^{25–109}$ was immobilized using the Biotin CAPture Kit, Series S (Cytiva) to ~130 RU. For each injection, Helicons in 10 mM DMSO stock were diluted to 10 μM in SPR running buffer and were injected over the surface for 120 s at 30 μL/min to achieve equilibrium binding. 150 nM tag-free CTNNB1$^{134–665}$ protein was then injected for 60 s at 30 μL/min in the absence or presence of Helicons over the surface. The chip surface was regenerated with a 120-second injection of CAP regeneration solution each cycle. Data were analyzed using Biacore Insight Evaluation software (Cytiva). Sensorgrams were double-referenced and evaluated for competition.

For kinetics experiments of the MDM2:β-catenin ternary complex, SPR ABA experiments were performed on a Biacore S200 (Cytiva)

instrument at 25 °C in 1x HBS-P+ buffer (Cytiva) with 1% DMSO and 0.5 mM TCEP. Biotinylated MDM2$^{25–109}$ was immobilized using the Biotin CAPture Kit, Series S (Cytiva) to ~160 RU. For each injection, Helicons in 10 mM DMSO stock were diluted to 10 μM in SPR running buffer and were injected over the surface for 120 s at 30 μL/min to achieve equilibrium binding. 2.5 μM tag-free CTNNB1$^{134–665}$ protein was serially diluted 1:3 for 8 total concentrations, then injected over the surface for 180 s at 30 μL/min in the presence of 10 μM Helicon. The chip surface was regenerated with a 120-second injection of CAP regeneration solution each cycle. Data were analyzed using Biacore Insight Evaluation software (Cytiva). Sensorgrams were double-referenced and fit to steady state affinity models to determine $K_D$ and $R_{max}$. The percent of Helicon-induced ternary complex was calculated by first taking the molecular weights of each component of the ternary complex to determine the expected response in RU. This value was then divided by the experimental response ($R_{max}$) and multiplied by one hundred to determine the percent of complex formed.

**SPR, ABA mode: PPIA: CHIP.** For kinetics experiments of the PPIA: CHIP ternary complex, SPR ABA experiments were performed on a Biacore S200 (Cytiva) instrument at 25 °C in 1x HBS-P+ buffer (Cytiva) with 1% DMSO and 0.5 mM TCEP. Biotinylated PPIA was immobilized using the Biotin CAPture Kit, Series S (Cytiva) to ~300 RU. For each injection, Helicons in 10 mM DMSO stock were diluted to 10 μM in SPR running buffer and were injected over the surface for 120 s at 30 μL/min to achieve equilibrium binding. 2.5 μM tag-free CHIP protein was serially diluted 1:3 for 8 total concentrations, then injected over the surface for 180 s at 30 μL/min in the presence of 10 μM Helicon. The chip surface was regenerated with a 120-second injection of CAP regeneration solution each cycle. Data was analyzed using Biacore Insight Evaluation software (Cytiva). Sensorgrams were double-referenced and fit to steady state affinity models to determine $K_D$ and $R_{max}$. The percent of Helicon-induced ternary complex was calculated by first taking the molecular weights of each component of the ternary complex to determine the expected response in RU. This value was then divided by the experimental response ($R_{max}$) and multiplied by one hundred to determine the percent of complex formed.

### Auto-ubiquitiylation of WWP2 (ELISA)

E3LITE Customizable Ubiquitin Ligase Kit (LifeSensors, UC101) was used to assess the autoubiquitination activity of HECT domain of WWP2 (WWP2$^{HECT}$). The ELISAs were performed with steps following the manufacturer protocol, with all solutions freshly made before the start of the experiment and protein components carefully stored on ice before addition ($n = 3$). The concentration of the catalytic HECT domain was 50 nM. The Helicon inhibitors were used at a concentration of 10 μM, with DMSO as the negative control. Relative Luminescence Units (RLUs) were recorded with a GloMax™ Discover luminometer (Promega).

### Assessment of ternary complex formation with Time-Resolved Fluorescence Energy Transfer (TR-FRET) and Fluorescence Polarization (FP)

All TR-FRET and FP experiments were repeated at least 2 times, as technical replicates (e.g., tested at different date), or/and biological replicates (with different batches of proteins).

**TR-FRET analysis of the TEAD4: CHIP complex.** For the TR-FRET ternary complex formation of the TEAD4: CHIP pair, biotinylated recombinant TEAD4$^{217–434}$ was diluted to 100 nM, Alexa Fluor™ 488 labeled CHIP$^{23–303}$ was diluted to 150 nM and Terbium-labeled streptavidin (Cis-Bio) was diluted to 2 nM in assay buffer (10 mM HEPES pH 7.5, 150 mM NaCl, 0.05% Tween-20) in a final volume of 20 μL per well of a black 384-well plate (Costar). Compounds were serially diluted in 90% DMSO and 40 nL of compound (11-point three-fold Helicon

dilution series with top concentration = 20 μM) was added to the plate and the samples were incubated for 60 min at room temperature ($n$ = 2). FRET signal was determined using a LanthaScreen™ filter on a PheraStar (BMG Biotech) plate reader (Ex: 337 nm; Em$_1$: 490 nM; Em$_2$: 520 nM). The ratio of Em$_{520}$ to Em$_{490}$ was calculated and plotted against compound concentration. Resulting data were fit to a four-parameter dose-response curve with variable slope. The positive control chimeric compound, P325, Ac-LWWPDGSGSGGSPGQVPMRKRQLPASFWEEPR-NH$_2$, is a designed bi-functional molecule with its N-terminus adapted from the CHIPopt peptide that interacts with CHIP[47], and the C-terminus derived from the FAM181A fragment that interacts with TEAD4[48]. The curve for the positive control was fitted with a Biphasic Curve function in Prism 9 (GraphPad).

**TR-FRET analysis of the PPIA: CHIP complex.** For the TR-FRET ternary complex formation of the PPIA: CHIP pair, biotinylated recombinant full length PPIA was diluted to 100 nM, Alexa Fluor™ 488 labeled CHIP[23–303] was diluted to 150 nM and Terbium-labeled strep-tavidin (Cis-Bio) was diluted to 2.3 nM in assay buffer (10 mM HEPES pH 7.5, 150 mM NaCl, 0.05% Tween-20) in a final volume of 20 μL per well of a black 384-well plate (Costar). Compounds were serially diluted in 90% DMSO, and 40 nL of compound (11-point three-fold Helicon dilution series with top concentration = 20 μM) was added to the plate. To probe the trimerizer interface of PPIA, CHIP, 2 μM CHIP-binding Helicon H318 or 2 μM cyclosporine A (CsA) was added to the assay buffer. Assay plates ($n$ = 2) were incubated for 60 min at room temperature. FRET signal was determined using a LanthaScreen™ filter on a PheraStar (BMG Biotech) plate reader (Ex: 337 nm; Em$_1$: 490 nM; Em$_2$: 520 nM). The ratio of Em$_{520}$ to Em$_{490}$ was calculated and plotted against compound concentration. The resulting data were fitted to a four-parameter dose-response curve with variable slope.

**FP analysis of the β-catenin: MDM2 and β-catenin:MDM4 complexes.** For the FP analysis of β-catenin ternary complex formation, compounds at 10 mM in DMSO were serially diluted 1:3 in DMSO for a total of 11 concentrations using a Mosquito LV (SPT Labtech), then diluted 1000-fold in buffer (50 mM HEPES, pH 7.5, 125 mM NaCl, 2% glycerol, 0.5 mM EDTA, 0.05% v/v pluronic acid) in duplicate by the Mosquito LV (SPT Labtech) into a black polystyrene 384-well plate (Corning) (11-point three-fold Helicon dilution series with top concentration = 10 μM). Protein-probe solution includes 80 nM full-length β-catenin or 130 nM β-catenin residues 134–665, mixed with 25 nM MDM2[17–111] or MDM4[14–111] labeled with Alexa Fluor™ 488. Protein-probe solution was plated into the Helicon plate using the MultiDrop Combi (Thermo Fisher) for a total reaction pool of 40 μL. The plate was incubated and protected from light for 1 h at room temperature prior to reading. ($n$ = 2) reads were performed on a CLARIOstar plate reader (BMG Labtech) with excitation at 485 nm, emission at 525 nm, and cutoff at 504 nm, and the resulting data were fit to a four-parameter dose-response curve with variable slope. To probe the trimerizer interface of MDM2:β-catenin, 2 μM ALRN-6924 peptide[15] or 100–400 nM ICAT (residues 1–81) recombinant proteins were added to the assay buffer, and the plates were prepared similarly. Positive control Helicon, P335, Ac-PD-cyclopentylalanine-CDDAAFNC-3Thi-benzothienylalanine-QGSGS-bAla-LTFEHYWAQLTS-NH$_2$ (Cys-stapled), is a designed bi-functional molecule consisting at its N-terminus of a β-catenin-interacting Helicon[22] and at its C-terminus, a p53-derived peptide that interacts with MDM2 (ref. 66). The curve for the positive control was fitted with the Biphasic Curve function in Prism 9.

**Competition fluorescence polarization**
**Competition FP of TEAD4.** For the competition FP of TEAD4, Helicons at 10 mM in DMSO were serially diluted 1:3 in DMSO for a total of 11 concentrations using a Mosquito LV (SPT Labtech), then diluted 1000-fold in buffer (50 mM HEPES, pH 7.5, 150 mM NaCl, 0.5 mM

TCEP, 0.05% v/v pluronic acid) in duplicate by the Mosquito LV (SPT Labtech) into a black polystyrene 384-well plate (Corning) (11-point three-fold Helicon dilution series with top concentration = 10 μM). 5FAM-labeled YAP residues 50–100 (Uniprot ID: P46937 with M73nLeu, M86nLeu for stability) was made synthetically by New England Peptide (Gardner, MA) by standard methods. Probe solution (40 nM TEAD4, mixed with 10 nM 5FAM-labeled YAP probe, in buffer) and plated using the MultiDrop Combi (Thermo Fisher) for a total reaction pool of 40 μL. The plate was incubated and protected from light for 1 h at room temperature prior to reading. Reads were performed on a CLARIOstar plate reader (BMG Labtech) with excitation at 485 nm, emission at 525 nm, and cutoff at 504 nm. Data were fitted to a 1:1 binding model with Hill slope using an in-house script.

**Competition FP of CHIP.** For the competition FP of CHIP, the assay was performed similarly to the competition FP of TEAD4. The assay was performed in buffer: 1 x HBS-P+ (Cytiva), with 400 nM CHIP[23–303] recombinant protein as target, and 20 nM CHIP-binding peptide as probe (5FAM-bAla-SSGPTIEEVD, derived from HSP70 (ref. 67). The plates were incubated and protected from light for 1 h at room temperature, and resulting data were fitted to a 1:1 binding model with Hill slope using an in-house script.

**Competition FP of MDM2.** For the competition FP of MDM2, the assay was performed similarly to the competition FP of TEAD4. The assay was performed in buffer: 1x PBS with 0.01% Tween, with 30 nM MDM2[17–111] recombinant protein as target, and 3 nM MDM2-binding peptide as probe (5FAM-bAla-LTFEHYWAQLTS-NH$_2$, derived from p53 (ref. 66)). The plates were incubated and protected from light for 1 h at room temperature, and resulting data were fitted to a 1:1 binding model with Hill slope using an in-house script.

**Competition FP of VHL.** For the competition FP of VHL, the assay was performed similarly to the competition FP of TEAD4. The assay was performed in the buffer: 10 mM HEPES, pH 7.5, 50 mM NaCl, 0.05% v/v pluronic acid, 15 nM VHL-ELOBC recombinant protein, and 5 nM VHL tracer, HXC78 (ref. 40). A structurally similar small molecule VHL binder, VH298 (Sigma SML1896)[41] was included in the competition FP assay as a positive control. The plates were incubated and protected from light for 1 h at room temperature, and resulting data were fitted to a 1:1 binding model with Hill slope using an in-house script. HXC78 systematic name: (2S,4R)-N-((S)−1-(3′,6′-dihydroxy-3-oxo-3H-spiro[isobenzofuran-1,9′-xanthen]−5-yl)−17-    (4-(4-methyl-thiazol-5-yl)phenyl)−1,15-dioxo-5,8,11-trioxa-2,14-diazaheptadecan-17-yl)−4-    hydroxy-1-((R)−3-methyl-2-(3-methylisoxazol-5-yl)buta-noyl)pyrrolidine-2-carboxamide; VHL298 systematic name: (2S,4R)−1-((S)−2-(1-Cyanocyclopropanecarboxamido)−3,3-dimethylbuta-noyl)−4-hydroxy-N-(4-(4-methylthiazol-5-yl)benzyl)pyrrolidine-2-carboxamide.

**Crystal structure data collection**
To obtain the structures of the protein-Helicon complexes, briefly, 10 mM of Helicon stock in 90% DMSO were added to the protein stocks to a final 1:1.25 protein: Helicon molar ratio or 1:1:1.25 protein-A: protein-B: Helicon and screened against commercially available crystallization screens. Crystals were obtained by hanging or sitting hanging drop vapor diffusion methods at room temperature, with their crystallization conditions detailed in Supplementary Data 2. Crystals were cryo-protected with glycerol or ethylene glycol, fol-lowed by flash-freezing in liquid nitrogen. Diffraction datasets were collected at 100 K at various sources as described and acknowledged in Supplementary Data 2. Data were processed in XDS[68] & XSCALE[69], AIMLESS[70], and/or STARANISO[71], all parts of the autoPROC suite[72]. Molecular replacement solutions were obtained using PHASER[73] with

previously deposited high-resolution PDB structures as search models, and we have indicated the starting models for each of the structures in the header of the mmcif and pdb coordinates. Complete models were built through iterative cycles of manual model building in COOT[74] and structure refinement was carried out using either REFMAC[75] or PHENIX[76], with all final refinements using PHENIX. All the structure model figures in the paper were prepared using PyMOL (The PyMOL Molecular Graphics System, Version 2.4, Schrödinger, LLC.). The atomic coordinates and structure factors have been deposited in the Protein Data Bank, www.pdb.org.

### Reporting summary

Further information on research design is available in the Nature Portfolio Reporting Summary linked to this article.

## Data availability

Atomic coordinates and structure factors have been deposited in the Protein Data Bank with accession codes 8EI9, 8EIA, 8EIB, 8EIC, 8EHZ, 8EI0, 8EI1, 8EI2, 8EI3, 8EI4, 8EI5, 8EI6, 8EI7, and 8EI8. The DNA sequencing data acquired in the screening efforts have been deposited in the NCBI Sequence Read Archive under accession code PRJNA1019768. All data needed to evaluate the conclusions of the study are present in the paper and the Supplementary files. Source data are provided with this paper. Requests for materials should be addressed to jmcgee@fogpharma.com. Source data are provided with this paper.

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

## Acknowledgements

We would like to thank M. Bucci for contributions to manuscript preparation, S. Brady for E3 ligase gene expression analysis, N. Lavey for assistance with biochemical assays, and S. Tong, W. Zhang, and P. Zhang at Viva Biotech for supervision of protein production and collection of X-ray diffraction data. Please refer to Supplementary Data 2 for detailed acknowledgments for the X-ray crystallography data collection.

## Author contributions

O.S.T., K.L., T.M.T., J.-M.S., G.L.V., and J.H.M. designed the study, O.S.T., K.L., T.L.T., J.-M.S., M.M., J.D.R., and S.L. generated reagents, O.S.T., K.L., T.L.T., J.-M.S., M.M., J.D.R., and S.L. performed experiments, O.S.T., K.L., T.L.T., T.M.T, J.-M.S., J.D.R., S.L., G.L.V., and J.H.M. analyzed data, O.S.T., K.L., and J.H.M. wrote the manuscript, and all authors edited the manuscript.

## Competing interests

The authors declare a competing interest. FOG Pharmaceuticals Inc. has a pending patent application that covers the peptide sequences

described in this manuscript, and methods for generating and screening trimerizer libraries (Provisional application number: 64/453,464). O.S.T., K.L., T.L.T., J.-M.S., G.L.V., J.H.M., and T.M.T. are inventors on the patent. O.S.T., T.L.T., T.M.T., J.D.R., S.L., and J.H.M. are currently employed by FOG Pharmaceuticals Inc., and G.L.V. serves on the board of directors of FOG Pharmaceuticals Inc. K.L. was an employee of FOG Pharmaceuticals Inc. and is currently employed by Kymera Therapeutics, Inc. J.-M.S. was an employee of FOG Pharmaceuticals Inc. and is currently employed by Relay Therapeutics, Inc. M.M. is a former employee of FOG Pharmaceuticals Inc.
