## [Peer Review File · Nature Communications]

Recognition and reprogramming of E3 ubiquitin ligase surfaces by α -helical peptidesEditorial Note: This manuscript has been previously reviewed at another journal that is not operating a transparent peer review scheme. This document only contains reviewer comments and rebuttal letters for versions considered at *Nature Communications* .

REVIEWERS' COMMENTS

Reviewer #1 (Remarks to the Author):

My main criticism was that for many graphs, only changes in activity were shown but not the absolute values (or the raw data), which made it impossible to see how substantial the changes were. The latter is important to tell if an assay is working efficiently and also to tell if the data may be prone to small variations/errors in the assay (i.e. small differences of large absolute values).

The authors have now provided the raw data for many of the graphs, which allowed me critically review some of the core data and its robustness. For many of the graphs, the raw data shows large changes (e.g. TR-FRET, FP), which is very good and indicates that changes seen are substantial. However, there are still some problems with the data presentation, as described below for some examples. I recommend i) to address the specific points below, ii) to check that the same problems are not found for other figure (I did not look at all!) and correct if necessary.

As indicated before, I like the work, in particular the concept and the impressive engineering effort, and can recommend publication if the points criticized can be addressed.

1. Labeling of axes

Many of the graphs are not properly labeled and it is often not clear how the values on the y-axes were calculated (in main figure, SI figure and in extended data figures). It is important to indicate "what was measured", the "unit" and in the figure caption how the values were calculated. For example Figure 4A, some graphs show "RFU" and it is not clear what this is exactly (one would assume some arbitrary units but it is not clear what). Or some show "RFU (Normalized, %)" but it is not clear what normalized mean (to which value it was normalized). For Figure 4c, the same applied (although there, I could guess what it means: most likely change relative to no helicon, where 100% is the initial FP?). The same problem applies in Figure 5b, and to several extended figures.

2. Different units for the same type of data

In Figure 4A, some graphs have the y-axes labeled with "RFU" and some with "RFU (Normalized, %)" although the type of experiment is the same. Why? It seems to me that the graph with "RFU" shows the absolute values. But why are these so different compared to raw data shown in SI Figure 1 (values from 0 to 3 vs. values from 0 to 50,000)?

3. Hook effect shown in Figure 4A

The raw data presented now in SI Figure shows a completely different picture than one would get when looking at main Figure 4A: The gray curve of the chimeric control shows a very small change (10% change) while the two curves of the helicons show large changes (800% change). This needs mentioning in the main text and critical discussion.

Reviewer #3 (Remarks to the Author):

As stated in my previous review, this is technically an excellent piece of work. I also consider it well worth publishing in Nature Comms. However, my key comments were not addressed, and were largely considered as out of scope by the authors. The current methodology does not necessarily allow to

strictly "reprogram" E3 ligases changing their substrate specificity (s. implicit claims in title, abstract, introduction, discussion and throughout). Instead, the methodology allows to generate binders/timerisers/helicons (which is arguably beautiful and important); yet in the cases shown these helicons did not support ubiquitination of the targets. There is no reason why they could not principally be targeting these neo-substrates if the sites of the helicons is compatible with ubiquitination! Yet, the approach is site agnostic and the best binders happened to be non-productive for ubiquitination. Hence the authors appear to generate neutral binders, or E3 inhibitors, rather than a way to reprogram specificity. In my opinion the paper needs to be substantially re-written, in vitro ubiquitination assays included even if negative, and the paper framed differently, such that the claims reflect the experiments.

Reviewer #4 (Remarks to the Author):

My concerns have been addressed. The supplementary excel sheet with the crystallographic data needs to be updated.

NCOMMS-23-35335-T
REVIEWERS' COMMENTS

Reviewer #1 (Remarks to the Author):

My main criticism was that for many graphs, only changes in activity were shown but not the absolute values (or the raw data), which made it impossible to see how substantial the changes were. The latter is important to tell if an assay is working efficiently and also to tell if the data may be prone to small variations/errors in the assay (i.e. small differences of large absolute values).

The authors have now provided the raw data for many of the graphs, which allowed me critically review some of the core data and its robustness. For many of the graphs, the raw data shows large changes (e.g. TR-FRET, FP), which is very good and indicates that changes seen are substantial. However, there are still some problems with the data presentation, as described below for some examples. I recommend i) to address the specific points below, ii) to check that the same problems are not found for other figure (I did not look at all!) and correct if necessary.

As indicated before, I like the work, in particular the concept and the impressive engineering effort, and can recommend publication if the points criticized can be addressed.

1. Labeling of axes

Many of the graphs are not properly labeled and it is often not clear how the values on the y-axes were calculated (in main figure, SI figure and in extended data figures). It is important to indicate “what was measured”, the “unit” and in the figure caption how the values were calculated. For example Figure 4A, some graphs show “RFU” and it is not clear what this is exactly (one would assume some arbitrary units but it is not clear what). Or some show “RFU (Normalized, %)” but it is not clear what normalized mean (to which value it was normalized). For Figure 4c, the same applied (although there, I could guess what it means: most likely change relative to no helicon, where 100% is the initial FP?). The same problem applies in Figure 5b, and to several extended figures.

For all graphs presented in the manuscript, we now include the raw data (in the Source Data file) as well as accompanying graphs of non-normalized data, including for the specific instances noted by the referee here. Specifically, these non-normalized data were used to generate graphs shown in Supplementary Figure 8 to support the graphs of normalized data shown in Figures 4a, 4c, and Supplementary Figures 2b and 4a-d.

We have also described more clearly in the figures and legends what exactly was measured (e.g. Emission ratio 520 nm/490 nm for TR-FRET data) and how normalization was done. We show normalized data when the differences in the magnitude of responses are great enough to make displaying them in the same graph difficult (e.g. for Figure 4A noted below by the referee).

2. Different units for the same type of data

In Figure 4A, some graphs have the y-axes labeled with “RFU” and some with “RFU (Normalized, %)” although the type of experiment is the same. Why? It seems to me that the graph with “RFU” shows the absolute values. But why are these so different compared to raw data shown in SI Figure 1 (values from 0 to 3 vs. values from 0 to 50,000)?

Thank you for the advice to better present the data in Figure 4A and all figures of biochemical data. As noted for point 1, we now use more precise units when displaying all raw data and describe better for all graphs how the normalized data were generated from the raw data.

3. Hook effect shown in Figure 4A

The raw data presented now in SI Figure shows a completely different picture than one would get when looking at main Figure 4A: The gray curve of the chimeric control shows a very small change (10% change) while the two curves of the helicons show large changes (800% change). This needs mentioning in the main text and critical discussion.

Indeed, the difference in scales for the experimental and control (raw) data compresses the latter (grey curve) significantly as shown in the left-most panel of Supplementary Fig. 8b. The graph on the right shows the scale to best visualize the behavior of the control:

This is why we chose to normalize the data for presentation in Fig. 4a:

We make note of this when describing the non-normalized data for Fig. 4a and Supplementary Fig. 8b (both in the text and in the figure legends).

Reviewer #3 (Remarks to the Author):

As stated in my previous review, this is technically an excellent piece of work. I also consider it well worth publishing in Nature Comms. However, my key comments were not addressed, and were largely considered as out of scope by the authors. The current methodology does not necessarily allow to strictly "reprogram" E3 ligases changing their substrate specificity (s. implicit claims in title, abstract, introduction, discussion and throughout). Instead, the methodology allows to generate binders/timerisers/helicons (which is arguably beautiful and important); yet in the cases shown these helicons did not support ubiquitination of the targets. There is no reason why they could not principally

be targeting these neo-substrates if the sites of the helicons is compatible with ubiquitination! Yet, the approach is site agnostic and the best binders happened to be non-productive for ubiquitination. Hence the authors appear to generate neutral binders, or E3 inhibitors, rather than a way to reprogram specificity. In my opinion the paper needs to be substantially re-written, in vitro ubiquitination assays included even if negative, and the paper framed differently, such that the claims reflect the experiments.

We thoroughly reviewed the title and text of the manuscript and were mindful to only suggest 'reprogramming' when describing the surface binding properties of the E3 ligases, and not their catalytic activity or specificity. We intend only to highlight that the Helicons can induce the E3s to bind neo-targets, such as the MDM2- β -catenin interactions we revealed in our structural analyses of the trimerizers. We also added additional language in the main text to reiterate that this reprogramming of binding - the induced proximity event - is the first step towards developing the types of tools that could be used for productive applications such as targeted protein degradation.

Also towards this point, during the course of our experiments, we did not attempt to collect conclusive data showing either ubiquitylation or lack of ubiquitylation. Due to this lack of data, we therefore do not make any claims one way or another about ubiquitylation in our manuscript, and thus felt it was most appropriate to refrain from presenting any preliminary or incomplete results that could be distracting or misleading. We instead chose to focus the manuscript on reporting our development of the phage screening platform for discovering Helicons that promote cooperative ternary complexes between E3 ligases and neo-target proteins, and, as noted above, to emphasize that further optimization and assessment of functional activity will be an important component of future studies.

Reviewer #4 (Remarks to the Author):

My concerns have been addressed. The supplementary excel sheet with the crystallographic data needs to be updated.

We have updated the crystallographic data tables in Supplementary Data 2 to reflect the changes the referee requested in their original report.